# Partial Diffusion Suffices: Solving General Inverse Problems via Score Evolution

## Abstract

Despite rapid advancements, pre-trained diffusion models still face dual challenges in solving inverse problems. On the one hand, current methods depend heavily on iterative optimization of the complete reverse diffusion chain, leading to difficulties in balancing manifold preservation and measurement constraints, especially under complex scenarios. On the other hand, strong assumptions about noise characteristics are necessitated, limiting their applicability in most practices. With the signal-noise variation of the diffusion score theoretically analyzed, this study proposes a plug-in alternative, dubbed Partial Diffusion with Score Evolution (PDSE), which optimizes the solution space by reconstructing a partial iterative path of the reverse diffusion flow. First, with the signal-noise equilibrium point of the score function derived, the critical diffusion steps are precisely identified, which preserve essential generative capabilities while controlling the propagation range of evolving errors. Second, by employing a standardized optimization approach, the iterative programme could suppress interference from noise of varying types and intensities. Extensive experiments across diverse scenarios demonstrate that PDSE significantly enhances adherence to measurement constraints while ensuring manifold feasibility, particularly exhibiting robust performance in nonlinear inverse scenarios with unknown noise characteristics. The aonoymous code is available on https://anonymous.4open.science/r/code-of-PDSE.

## 1 Introduction

Diffusion models implicitly capture the prior distribution of given data by learning the logarithmic density gradient $\nabla_{\boldsymbol{x}} \log p(\boldsymbol{x})$, which can be further leveraged to solve massive inverse problems Chung et al. (2023b). Mathematically, the goal is to recover an image $\boldsymbol{x} \in \mathbb{R}^d$ from noisy observations, i.e., $\boldsymbol{y} = \mathcal{A}\boldsymbol{x} + \boldsymbol{n}$, with $\boldsymbol{n} \sim \mathcal{N}(0, \sigma_y^2 I_{k \times d})$. Here, $\mathcal{A} \in \mathbb{R}^k$ denotes a known measurement operator, and $\boldsymbol{y} \in \mathbb{R}^k$ is the observation with an unknown noise variance $\sigma_y^2$. Given the forward model, sampling from the posterior distribution $p(x|y)$ can be achieved by incorporating the log-likelihood gradient, $\nabla_x \log p(y|x)$. However, despite the clear intuition, the likelihood term $p_t(y|x_t)$ is only tractable at $t = 0$, and becomes inaccessible for any $t > 0$, making analytical handling highly challenging Song et al. (2024); Rout et al. (2024).

To address this issue, several studies have attempted to approximate the measurement-guided score $\nabla \log p_t(y|x_t)$ via closed-form expressions Chung et al. (2023b); Wang et al. (2022); Kawar et al. (2022); Zhu et al. (2023). Practical manipulations include error projection estimation Chung et al. (2023b) and error-guided scaling Holzschuh et al. (2023). Nevertheless, these methods exhibit limited fidelity in complex scenarios, lacking robustness to noise variations and generalization to unseen degradations. An alternative line of work employs variational inference to approximate the true posterior $p(x|y)$ using a simpler and more tractable distribution, whose parameters are optimized through variational objectives Mardani et al. (2024); Vargas & Nowozin (2023); Alkan et al. (2023); Feng et al. (2023). These methods reformulate posterior modeling as an optimization task, thereby reducing computational complexity and making them suitable for high-dimensional and nonlinear inverse problems. However, they suffer from suboptimal solutions and sensitivity to the choice of variational family Alkan et al. (2023), which may introduce bias and degrade reconstruction quality.

Beyond the above approaches, more recent studies have attempted optimizing initial noise in the latent space of diffusion models through backpropagation Wang et al. (2024); Daras et al. (2022);

Holzschuh et al. (2023), which can be considered as a generalization of the classical Conditional Score-based Generative Model (CSGM) to the diffusion framework. In these attempts, one searches for an optimal latent vector $\boldsymbol{z}^* = \arg\min_{\boldsymbol{z}} \|\boldsymbol{y} - \mathcal{A}\hat{\boldsymbol{x}}(\boldsymbol{z})\|_2^2$, with $\hat{\boldsymbol{x}}(\boldsymbol{z})$ being the generated signal. However, solving for $\boldsymbol{z}^*$ requires numerous gradient descent iterations, resulting in slow convergence. Moreover, the concerned backpropagation incurs exponentially growing memory costs, often limiting the sampling steps, which may in turn lead to biased generation. In brief, investigations within latent-space optimization in diffusion models demand further exploration.

To fill up the gap, we propose a new solution, namely *Partial Diffusion with Score Evolution (PDSE)*, for general inverse problems, optimizing posterior approximation in CSGM-based approaches. By examining the variation trend of the score norm $\|\mathbf{s}(\boldsymbol{x}_t, t)\|_2$, we perform a evolution analysis and identify a critical timestep $t^*$. On that basis, we argue that partial diffusion suffices, starting the process at step $t^*$ that balances well the signal and noise while preserving the core data manifold structure. More importantly, most of the redundant sampling steps are naturally eliminated. During optimization, the intermediate variable $\boldsymbol{x}_{t^*}$ is updated via backpropagation, integrated with a scheme of Adaptive Learning and Early Stopping (ALES), which further improves reconstruction stability and robustness under unknown degradations. Unlike traditional variational inference, PDSE approximates the posterior $p(x|y)$ without parametric assumptions, avoiding distribution-induced bias. Moreover, PDSE is equipped with a measurement-consistency gradient correction near $t^*$, which accelerates convergence while maintaining manifold feasibility. The final proposal plays in a plug-and-play (PnP) manner for most noisy inverse problems, such as image inpainting, super-resolution, deblurring, and Fourier phase retrieval, enjoying superior performance to current approaches.

## 2 BACKGROUNDS

### 2.1 PRELIMINARIES OF DIFFUSION MODELS

The forward diffusion process gradually introduces Gaussian noise into data $\boldsymbol{x}_0 \sim p_{\text{data}}$ through the following Stochastic Differential Equation (SDE):

$$d\boldsymbol{x} = -\frac{\beta(t)}{2}\boldsymbol{x}dt + \sqrt{\beta(t)}d\boldsymbol{w}, \quad t \in [0, T], \tag{1}$$

where $\beta(t) : [0, T] \to \mathbb{R}^+$ controls the noise schedule, and $d\boldsymbol{w}$ denotes Wiener process increments. The corresponding reverse-time SDE:

$$d\boldsymbol{x} = \left[ -\frac{\beta(t)}{2}\boldsymbol{x} - \beta(t) \underbrace{\nabla_{\boldsymbol{x}_t} \log p_t(\boldsymbol{x}_t)}_{\varepsilon_\theta(\boldsymbol{x}_t, t)} \right] dt + \sqrt{\beta(t)}\, d\bar{\boldsymbol{w}}, \tag{2}$$

reveals generation as a stochastic decoding process guided by the score function $\nabla_{\boldsymbol{x}} \log p_t(\boldsymbol{x})$, which is approximated by a neural network $\varepsilon_\theta(\boldsymbol{x}_t, t)$ through denoising score matching Vincent (2011). Note that $d\bar{\boldsymbol{w}}$ refers to the Wiener process running in reverse time. The Denoising Diffusion Probabilistic Model (DDPM) Ho et al. (2020) discretizes this framework via Markovian transitions, using discrete noise levels $\beta_t$ sampled from the continuous schedule $\beta(t)$:

$$\boldsymbol{x}_t = \sqrt{1 - \beta_t}\boldsymbol{x}_{t-1} + \sqrt{\beta_t}\boldsymbol{z} \quad \text{(Forward)}, \tag{3}$$

$$\boldsymbol{x}_{t-1} = \frac{1}{\sqrt{\alpha_t}}\boldsymbol{x}_t - \frac{\beta_t}{\sqrt{1-\bar{\alpha}_t}}\varepsilon_\theta(\boldsymbol{x}_t) + \sqrt{\beta_t}\boldsymbol{z} \quad \text{(Reverse)}, \tag{4}$$

where $\alpha_t := 1 - \beta_t$, $\bar{\alpha}_t := \prod_{s=1}^t \alpha_s$, and $\boldsymbol{z} \sim \mathcal{N}(\boldsymbol{0}, \boldsymbol{I})$ is standard Gaussian noise. While theoretically sound, DDPM requires around $\mathcal{O}(10^3)$ steps due to strict adherence to SDE dynamics. Denoising Diffusion Implicit Models (DDIM) Song et al. (2021a) overcome this limitation through non-Markovian reparameterization, which employs Tweedie estimator to predict the clean signal:

$$\hat{\boldsymbol{x}}_0(\boldsymbol{x}_t) = \frac{\boldsymbol{x}_t - \sqrt{1 - \bar{\alpha}_t}\varepsilon_\theta(\boldsymbol{x}_t)}{\sqrt{\bar{\alpha}_t}}. \tag{5}$$

Then, the deterministic DDIM update becomes:

$$\boldsymbol{x}_{t-1} = \sqrt{\bar{\alpha}_{t-1}}\hat{\boldsymbol{x}}_0(\boldsymbol{x}_t) + \sqrt{1 - \bar{\alpha}_{t-1}}\varepsilon_\theta(\boldsymbol{x}_t, t). \tag{6}$$

By replacing the incremental stochastic updates of DDPM with deterministic manifold projections, DDIM achieves generation in around $\mathcal{O}(10^1)$ steps while guaranteeing high sample quality.

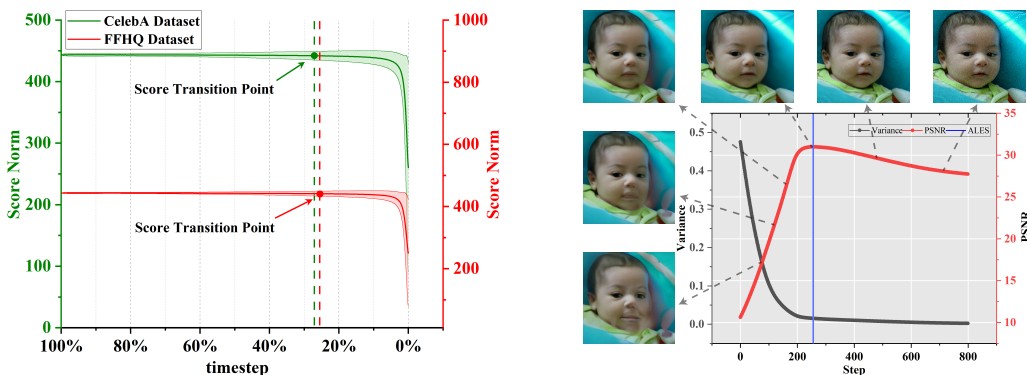

(a) Variation of $\|\varepsilon_\theta(x_t, t)\|_2$ during denoising      (b) PSNR and variance evolution over iterations

Figure 1: (Left) Evolution of the predicted noise norm $\|\varepsilon_\theta\|_2$. The transition point at 26% timestep motivates our partial diffusion strategy. (Right) PSNR and variance for our method, where the blue line indicates the ALES early stopping point, effectively halting training near the PSNR peak.

## 2.2 SOLVING INVERSE PROBLEMS WITH DIFFUSION MODELS

Diffusion models (DMs) for inverse problems (IPs) generally fall into two categories: sampling-based methods and optimization-based methods. The sampling-based branch aims to recover the original signal $\boldsymbol{x}$ from a noisy measurement $\boldsymbol{y} = \mathcal{A}(\boldsymbol{x}) + \boldsymbol{n}$ by iteratively approximating the conditional posterior $p(\boldsymbol{x}|\boldsymbol{y})$ through stochastic steps. These methods are mainly built on the reverse diffusion process, governed by the stochastic differential equation in Eq. (2), with the aid of some adaptations by incorporating measurement information. A common practice estimates the likelihood gradient $\nabla_{\boldsymbol{x}_t} \log p_t(\boldsymbol{y}|\boldsymbol{x}_t)$ using the Tweedie estimator in Eq. (5) Song et al. (2021b); Peng et al. (2024), followed by iterative corrections with a term like $\mathcal{A}^\top(\boldsymbol{y} - \mathcal{A}\hat{\boldsymbol{x}}_0)$. To speed up, some investigations refine this gradient as $\nabla_{\boldsymbol{x}_t} \log p_t(\boldsymbol{y}|\boldsymbol{x}_t) \approx \boldsymbol{J}_{\hat{\boldsymbol{x}}_0}^\top \frac{\mathcal{A}^\top(\boldsymbol{y} - \mathcal{A}\hat{\boldsymbol{x}}_0)}{\sigma_y^2}$, where $\boldsymbol{J}_{\hat{\boldsymbol{x}}_0} = \frac{\partial \hat{\boldsymbol{x}}_0}{\partial \boldsymbol{x}_t}$ is the Jacobian matrix Benton et al. (2023); Daras et al. (2024); Chung et al. (2023b). Others employ Langevin dynamics for updates Song et al. (2021a); Peng et al. (2024); Zheng et al. (2025), which are expressed as follows:

$$\boldsymbol{x}_t^{k+1} = \boldsymbol{x}_t^k + \eta[\varepsilon_\theta(\boldsymbol{x}_t^k, t) + \frac{\beta(t)\mathcal{A}^\top(\boldsymbol{y} - \mathcal{A}\boldsymbol{x}_t^k)}{2(\sigma_y^2 + \sigma_t^2)}] + \sqrt{2\eta}\boldsymbol{z}_t^k. \tag{7}$$

The optimization-based methods, by contrast, pursue deterministic solutions to IPs. One general strategy lies in the optimization of an initial latent noise $\boldsymbol{x}_T$, so that the output $G_\theta(\boldsymbol{x}_T)$, generated via the reverse probability flow ODE $\frac{d\boldsymbol{x}_t}{dt} = -\frac{\beta(t)}{2}\boldsymbol{x}_t - \frac{\beta(t)}{2}\varepsilon_\theta(\boldsymbol{x}_t, t)$, satisfies $\min_{\boldsymbol{x}_T} \|\boldsymbol{y} - \mathcal{A}(G_\theta(\boldsymbol{x}_T))\|^2 + \lambda\|\boldsymbol{x}_T\|_{\mathcal{H}^{-1}}^2$, with the Sobolev norm curbing high-frequency components Kawar et al. (2022); Song et al. (2024); Zirvi et al. (2024); Chung et al. (2024); Wang et al. (2024). Another strategy attempts to construct a posterior $q_\phi(\boldsymbol{x})$ and tune it by optimizing the following function, as intensively investigated in Luo (2022); Vargas & Nowozin (2023); Rout et al. (2024):

$$\mathcal{L}(\phi) = \mathbb{E}_{q_\phi}[\|\boldsymbol{y} - \mathcal{A}\boldsymbol{x}_0\|^2 - \int_0^T \frac{\beta(t)}{2}\|\varepsilon_\theta(\boldsymbol{x}_t, t) - \nabla_{\boldsymbol{x}_t} \log q_\phi(\boldsymbol{x}_t)\|^2 dt]. \tag{8}$$

This treatment balances data fidelity $\|\boldsymbol{y} - \mathcal{A}\boldsymbol{x}_0\|^2$ with the learned diffusion prior, aligning the posterior to both the measurement and the model's generative knowledge. Beyond these, SDEdit Meng et al. (2021), DiffPure Nie et al. (2022), and FMPlug Wan et al. (2025) also work with partially noised states, mainly to shorten denoising or pick task-specific starting times along the diffusion chain. DiffPure refines samples along a reverse SDE path with adjoint gradients, whereas PDSE optimizes the latent at a model-dependent but task-agnostic critical timestep $t^*$ via a measurement-consistency loss, rather than propagating gradients through the full diffusion trajectory.

## 3 METHODOLOGY

### 3.1 SCORE EVOLUTION ANALYSIS

Recall that the score function $\mathbf{s}(\boldsymbol{x}_t, t) = \nabla_{\boldsymbol{x}_t} \log p(\boldsymbol{x}_t|\boldsymbol{x}_0)$ is defined as the gradient of the log-probability density with respect to the noisy input. For a variance-preserving diffusion process, where $\boldsymbol{x}_t = \sqrt{\bar{\alpha}_t}\boldsymbol{x}_0 + \sqrt{1-\bar{\alpha}_t}\epsilon$, the score is directly related to the network's noise prediction $\varepsilon_\theta(\boldsymbol{x}_t, t)$ via the equation:

$$\mathbf{s}(\boldsymbol{x}_t, t) = -\frac{\boldsymbol{x}_t - \sqrt{\bar{\alpha}_t}\boldsymbol{x}_0}{1-\bar{\alpha}_t} \approx -\frac{\varepsilon_\theta(\boldsymbol{x}_t, t)}{\sqrt{1-\bar{\alpha}_t}}. \tag{9}$$

From this relationship, we can view the score function $\mathbf{s}(\boldsymbol{x}_t, t)$ as comprising two key components: a **directional guidance** provided by the neural network's output $-\varepsilon_\theta$, and a **time-dependent magnitude scaling** factor $1/\sqrt{1-\bar{\alpha}_t}$. While the theoretical norm of the full score, $||\mathbf{s}(\boldsymbol{x}_t, t)||$, increases as noise decreases due to the scaling factor, the evolution of the core generative information is better captured by analyzing the norm of the network's direct output, $||\varepsilon_\theta(\boldsymbol{x}_t, t)||$.

The norm of this predicted noise characterizes the evolution of effective generative guidance, as Fig. 1a demonstrates its monotonic decreasing behavior during the denoising process. Upon examining the figure, we observe that the initial phase exhibits a slow decay, after which the norm drops more rapidly. Unfortunately, this slow decay phase occupies a significant portion of the denoising process, implying potential redundancy within the early evolving steps. This raises a natural question: **Can we identify a critical threshold $t^*$, beyond which redundant denoising steps are minimized while retaining the diffusion model's effectiveness?** We explore this idea by formalizing $t^*$ in Proposition 3.1, which optimizes the denoising benefits.

**Proposition 3.1** (Score Transition Threshold). *In variance preserving diffusion models, there exists a unique critical time $t^*$ where:*

1. *The signal-to-noise ratio* $\mathrm{SNR}(t^*) = \frac{\mathrm{Var}(\sqrt{\bar{\alpha}_{t^*}}\boldsymbol{x}_0)}{\mathrm{Var}(\sqrt{1-\bar{\alpha}_{t^*}}\epsilon)} = \frac{\bar{\alpha}_{t^*}}{1-\bar{\alpha}_{t^*}} = 1$,

2. *The critical time $t^*$ satisfies the condition $\int_0^{t^*} \beta(s)\,ds = \ln 2$, which determines $t^*$ based on the accumulated variance.*

From this, we derive the value of $\bar{\alpha}_{t^*}$, representing the fraction of signal remaining at $t^*$:

$$\frac{\bar{\alpha}_{t^*}}{1-\bar{\alpha}_{t^*}} = 1 \implies \bar{\alpha}_{t^*} = \frac{1}{2}. \tag{10}$$

In practice, a continuous schedule $\bar{\alpha}_t = \exp\left(-\int_0^t \beta(s)\,ds\right)$ gives $\int_0^{t^*} \beta(s)\,ds = \ln 2$. And adopting DDPM's discrete linear schedule yields $t^* \approx 0.26T$. More detailed proofs and explanations are in Appendix B.1.

### 3.2 PLUG-IN STRATEGY WITH DIFFUSION PRIOR

Given the critical point $t^*$ where $\bar{\alpha}_{t^*} = \frac{1}{2}$, we define the reverse denoising process as

$$R(\cdot) = g_{\epsilon_\theta^{(0)}} \circ \cdots \circ g_{\epsilon_\theta^{(t^*)}}, \tag{11}$$

where $g_{\epsilon_\theta^{(i)}}$ maps $\boldsymbol{x}_{i+1}$ to $\boldsymbol{x}_i$ using the noise prediction $\epsilon_\theta^{(i)}$. This enables the diffusion process to serve as a prior for inverse problems, reconstructing signals from degraded observations by leveraging its generative strength. Another central question then emerges: **how much generative capacity is retained when starting the reverse process at $t^*$?** We explore this with the following proposition.

**Proposition 3.2** (Partial Generation Fidelity Bound). *Let $R(\cdot)$ denote the reverse process starting from $t^*$, and let $R_{full}(\cdot)$ be the full reverse process from $t = T$. Under the assumption that $\bar{\alpha}_{t^*} = \frac{1}{2}$, the KL divergence between the generated distribution and the data distribution satisfies:*

$$D_{KL}(p_{data}\|p_{R(x_{t^*})}) \leq \frac{\ln 2}{\int_0^T \beta(s)\,ds} \cdot D_{KL}(p_{data}\|p_{R_{full}(x_T)}). \tag{12}$$

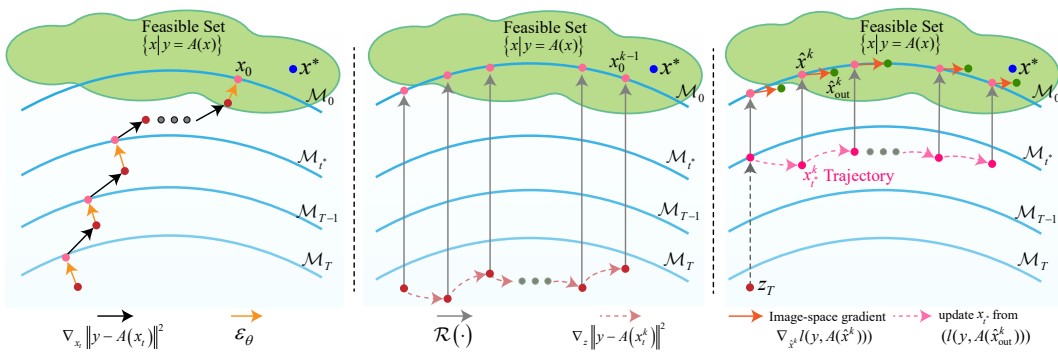

Figure 2: Comparisons on different inverse problem solvers based on diffusion models. (Left) Interleaving methods that alternate between diffusion steps and measurement consistency updates; (Middle) Previous plug-in method that performs full diffusion process, with $R(\cdot)$ at each iteration; (Right) Our solution that optimizes near the critical timestep $t^*$ and employs a measurement-consistency gradient correction for faster convergence while maintaining manifold feasibility.

From this proposition, the partial process retains at least $1 - \frac{\ln 2}{\int_0^T \beta(s)\,ds}$ fraction of the full generative fidelity in terms of relative KL discrepancy. This practice reflects the nonlinear contribution of denoising steps. For example, under a linear noise schedule with $\int_0^T \beta(s)\,ds \approx 10$, the partial process achieves approximately 93% of the full fidelity. The detailed proof is given in Appendix B.2.

Leveraging the high fidelity at the critical timestep $t^*$, we initialize the latent variable $\boldsymbol{x}_{t^*}$ by running a guided reverse diffusion process, which is then optimized using a hybrid and accelerated iterative strategy to enhance convergence. As illustrated in Fig. 2 (Right), this strategy combines a *manual correction step* in the image space with an *autograd-based optimization step* in the latent space to update $\boldsymbol{x}_{t^*}$.

At each iteration $k$, the process begins by decoding the current latent code $\boldsymbol{x}_{t^*}$ into an image estimate, $\hat{\boldsymbol{x}}^k = R(\boldsymbol{x}_{t^*})$. A measurement-consistency correction is then performed directly in the image space by taking a gradient step on the image estimate with respect to the measurement loss:

$$\hat{\boldsymbol{x}}_{\text{out}}^k = \hat{\boldsymbol{x}}^k - \eta \nabla_{\hat{\boldsymbol{x}}^k} l(\boldsymbol{y}, \mathcal{A}(\hat{\boldsymbol{x}}^k)), \tag{13}$$

where $\eta$ is the step size for this correction. The main optimization objective is the loss computed on the corrected image, $l(\boldsymbol{y}, \mathcal{A}(\hat{\boldsymbol{x}}_{\text{out}}^k))$. The gradient of this objective is then backpropagated through the decoder $R(\cdot)$ to update the latent variable $\boldsymbol{x}_{t^*}$ using the Adam optimizer.

### 3.3 ROBUSTNESS TO UNKNOWN NOISE

For noisy inverse problems $\boldsymbol{y} = \mathcal{A}(x_0) + n$, using diffusion priors $R(\cdot)$ often leads to noise overfitting, known as the "early-learning-then-overfitting" (ELTO) phenomenon Wang et al. (2024), where quality peaks then declines (Fig. 1b). The ES-WMV method Wang et al. (2023) attempts to mitigate ELTO by stopping when its running variance metric,

$$\text{VAR}(t) = \frac{1}{W} \sum_{w=0}^{W-1} \left\| x_{t+w} - \frac{1}{W} \sum_{i=0}^{W-1} x_{t+i} \right\|_F^2, \tag{14}$$

stabilizes. However, ES-WMV faces limitations with diverse noises: the uniform weighting may miss recent trends, a fixed patience $P$ overlooks training stage variations, and the focus of global variance captures little local changes in critical regions.

To address these shortcomings, we propose a method called Adaptive Learning and Early Stopping(ALES). This approach incorporates three key enhancements: a time-weighted variance calculation that prioritizes recent iterations for improved sensitivity to optimization trends. A stage-aware adaptation mechanism that adjusts stopping patience based on the optimization phase to balance stability and responsiveness, and a local sensitivity detection component to better identify significant changes in critical image regions. These features enable ALES to determine the optimal stopping point more robustly. A detailed explanation of each component and the full implementation are

---

**Algorithm 1:** PDSE + ALES Update Strategy.

---

**Input:** Critical timestep $t^*$, measurement $\boldsymbol{y}$, initial noise $z$, step size $\eta$, maximum iterations $N$.
**Output:** Recovered image $\hat{\boldsymbol{x}}_{\text{out}}$.

1   $\boldsymbol{x}_T \leftarrow z, k \leftarrow 0$
2   $\boldsymbol{x}_{t^*} \leftarrow \text{sample}(\boldsymbol{x}_T, t^*, \boldsymbol{y})$
3   Initialize $\boldsymbol{x}_{t^*}$ and $\eta$ as trainable parameters
4   optimizer $\leftarrow \text{Adam}(\boldsymbol{x}_{t^*})$
5   **while** $k < N$ **and** *the ALES condition is not triggered* **do**
6      $\hat{\boldsymbol{x}} \leftarrow R(\boldsymbol{x}_{t^*})$
7      $\boldsymbol{g}_{\text{manual}} \leftarrow \nabla_{\hat{\boldsymbol{x}}} \|\boldsymbol{y} - \mathcal{A}(\hat{\boldsymbol{x}})\|^2$
8      optimizer.zero_grad()
9      $\hat{\boldsymbol{x}}_{\text{out}} \leftarrow \hat{\boldsymbol{x}} - \eta \boldsymbol{g}_{\text{manual}}$
10      loss $\leftarrow l(\boldsymbol{y}, \mathcal{A}(\hat{\boldsymbol{x}}_{\text{out}}))$
11      loss.backward()
12      optimizer.step()
13      Uses ALES for early stop check
14      $k \leftarrow k + 1$

15   **return** $\hat{\boldsymbol{x}}_{\text{out}}$.

---

| Methods | **FFHQ** (256 × 256) | | | | | | **CelebA** (256 × 256) | | | | | |
|---|---|---|---|---|---|---|---|---|---|---|---|---|
| | **Super-resolution** (4×) | | | **Inpainting** (Random 70%) | | | **Super-resolution** (4×) | | | **Inpainting** (Random 70%) | | |
| | PSNR ↑ | SSIM ↑ | LPIPS↓ | PSNR ↑ | SSIM ↑ | LPIPS ↓ | PSNR ↑ | SSIM ↑ | LPIPS ↓ | PSNR ↑ | SSIM ↑ | LPIPS ↓ |
| DPS (ICLR 2023) | 24.679 | 0.552 | 0.180 | 32.212 | 0.821 | 0.082 | 25.958 | 0.591 | 0.145 | _33.443_ | 0.841 | _0.067_ |
| MCG (NeurIPS 2022) | 18.219 | 0.218 | 0.771 | 30.658 | 0.689 | 0.086 | 19.967 | 0.265 | 0.713 | 32.649 | 0.810 | 0.091 |
| MPGD (ICLR 2024) | 26.854 | 0.622 | 0.113 | 26.589 | 0.709 | 0.141 | 28.391 | 0.672 | 0.108 | 27.306 | 0.712 | 0.163 |
| DiffPIR (CVPR 2023) | 28.312 | 0.743 | 0.109 | 30.127 | 0.857 | 0.121 | 27.244 | 0.758 | 0.083 | 31.092 | 0.862 | 0.098 |
| ReSample (ICLR 2024) | 28.133 | 0.711 | 0.104 | 31.215 | 0.849 | _0.065_ | 29.864 | 0.761 | _0.077_ | 32.859 | 0.876 | 0.083 |
| DMPlug (NeurIPS 2024) | _29.032_ | _0.746_ | _0.098_ | _32.331_ | _0.880_ | 0.076 | _30.681_ | _0.764_ | 0.127 | 33.017 | _0.848_ | 0.096 |
| PDSE (Ours) | **29.544** | **0.767** | **0.062** | **32.753** | **0.884** | **0.041** | **30.915** | **0.780** | **0.075** | **34.048** | **0.896** | **0.045** |
| Ours vs. Best compe. | ▲0.512 | ▲0.021 | ▼0.036 | ▲0.422 | ▲0.004 | ▼0.024 | ▲0.234 | ▲0.016 | ▼0.002 | ▲0.605 | ▲0.042 | ▼0.022 |

| Methods | **FFHQ** (256 × 256) | | | | | | **CelebA** (256 × 256) | | | | | |
|---|---|---|---|---|---|---|---|---|---|---|---|---|
| | **Gaussian Deblurring** | | | **Motion Deblurring** | | | **Gaussian Deblurring** | | | **Motion Deblurring** | | |
| | PSNR ↑ | SSIM ↑ | LPIPS ↓ | PSNR ↑ | SSIM ↑ | LPIPS ↓ | PSNR ↑ | SSIM ↑ | LPIPS ↓ | PSNR ↑ | SSIM ↑ | LPIPS ↓ |
| DPS (ICLR 2023) | 26.312 | 0.645 | 0.149 | 28.161 | 0.721 | 0.104 | 27.290 | 0.660 | 0.109 | 28.532 | 0.691 | 0.108 |
| MCG (NeurIPS 2022) | 10.419 | 0.032 | 1.017 | 11.274 | 0.058 | 0.727 | 11.116 | 0.023 | 0.739 | 11.020 | 0.015 | 0.689 |
| MPGD (ICLR 2024) | 26.269 | 0.607 | _0.136_ | 30.142 | _0.789_ | _0.072_ | 27.705 | 0.656 | 0.107 | 27.438 | 0.678 | 0.092 |
| DiffPIR (CVPR 2023) | 26.025 | 0.729 | 0.168 | - | - | - | 26.990 | 0.744 | 0.263 | - | - | - |
| ReSample (ICLR 2024) | 27.866 | _0.667_ | 0.153 | _30.547_ | 0.773 | 0.089 | _29.452_ | _0.732_ | _0.095_ | _32.413_ | _0.833_ | **0.089** |
| DMPlug (NeurIPS 2024) | _28.068_ | 0.666 | 0.186 | 28.221 | 0.686 | 0.168 | 27.201 | 0.564 | 0.266 | 26.468 | 0.570 | 0.237 |
| PDSE (Ours) | **29.328** | **0.752** | **0.128** | **31.276** | **0.812** | **0.062** | **30.290** | **0.762** | **0.088** | **33.151** | **0.837** | _0.091_ |
| Ours vs. Best compe. | ▲1.260 | ▲0.085 | ▼0.008 | ▲0.729 | ▲0.023 | ▼0.010 | ▲0.838 | ▲0.030 | ▼0.007 | ▲0.738 | ▲0.004 | ▲0.002 |

Table 1: **(Linear Inverse Problems)** **Super-resolution**, **Inpainting**, **Gaussian deblurring** and **Motion deblurring** with additive Gaussian noise ($\sigma = 0.01$) (**Bold**: Best performance, Underlined: Second best performance, Red: Performance increase, Green: Performance decrease).

provided in Appendix B.5 and Algorithm 2. As demonstrated by the blue line in Fig. 1b, ALES effectively halts iterations near the PSNR peak.

## 4   EXPERIMENTS AND ANALYSIS

This section benchmarks the proposed method against state-of-the-art (SOTA) techniques across four linear image processing problems: **super-resolution**, **inpainting**, **Gaussian blur**, **motion blur**, and **MRI reconstruction**, and two nonlinear applications, namely **nonlinear deblurring** and **blind image deblurring with turbulence (BID)**. Generally, Diffusion Posterior Sampling (DPS) Chung et al. (2023b) and Manifold Constrained Gradients (MCG) Chung et al. (2022) are employed as baselines. Besides, sampling-based methods including Manifold Preserving Guided Diffusion (MPGD) He et al. (2024) and ReSample Song et al. (2024), as well as off-the-shelf PnP-based approaches such as DiffPIR Zhu et al. (2023) and DMPlug Wang et al. (2024), are also selected as the competing methods. For the issue of BID with turbulence, we introduce BlindDPS Chung et al. (2023a) as a baseline for its tailored implementation. Following DMPlug, we sample 100 images from the CelebA Liu et al. (2015) and FFHQ Karras et al. (2019) datasets, resized to $256 \times 256$ pixels.

| (a) **Nonlinear deblurring** with additive Gaussian noise ($\sigma = 0.01$) | | | | | | |
|---|---|---|---|---|---|---|
| | **FFHQ** ($256 \times 256$) | | | **CelebA** ($256 \times 256$) | | |
| Methods | PSNR ↑ | SSIM ↑ | LPIPS ↓ | PSNR ↑ | SSIM ↑ | LPIPS ↓ |
| DPS (ICLR 2023) | 26.611 | 0.656 | 0.127 | 29.137 | 0.755 | 0.127 |
| MPGD (ICLR 2024) | 18.202 | 0.226 | 0.650 | 17.974 | 0.207 | 0.750 |
| ReSample (ICLR 2024) | 29.713 | 0.763 | 0.126 | 29.876 | 0.759 | 0.107 |
| DMPlug (NeurIPS 2024) | 28.371 | 0.745 | 0.135 | 28.581 | 0.739 | 0.165 |
| PDSE (Ours) | **31.116** | **0.823** | **0.110** | **32.766** | **0.877** | **0.082** |
| Ours vs. Best compe. | ▲1.403 | ▲0.060 | ▼0.016 | ▲2.890 | ▲0.118 | ▼0.025 |
| (b) **BID with turbulence** under additive Gaussian noise ($\sigma = 0.01$) | | | | | | |
| | **FFHQ** ($256 \times 256$) | | | **CelebA** ($256 \times 256$) | | |
| Methods | PSNR ↑ | SSIM ↑ | LPIPS ↓ | PSNR ↑ | SSIM ↑ | LPIPS ↓ |
| BlindDPS (CVPR 2023) | 26.258 | 0.731 | 0.168 | 26.656 | 0.675 | 0.146 |
| DMPlug (NeurIPS 2024) | 27.860 | 0.783 | 0.146 | 28.131 | 0.741 | 0.132 |
| PDSE (Ours) | **28.872** | **0.838** | **0.119** | **29.264** | **0.793** | **0.106** |
| Ours vs. Best compe. | ▲1.012 | ▲0.055 | ▼0.017 | ▲1.133 | ▲0.052 | ▼0.106 |

Table 2: **(Nonlinear Inverse Problems)** Nonlinear deblur (a) and BID with turbulence (b) under additive Gaussian noise ($\sigma = 0.01$); for task (b), the BlindDPS algorithm is introduced to enable better comparison. (**Bold**: Best performance, Underlined: Second best, Red: Performance increase).

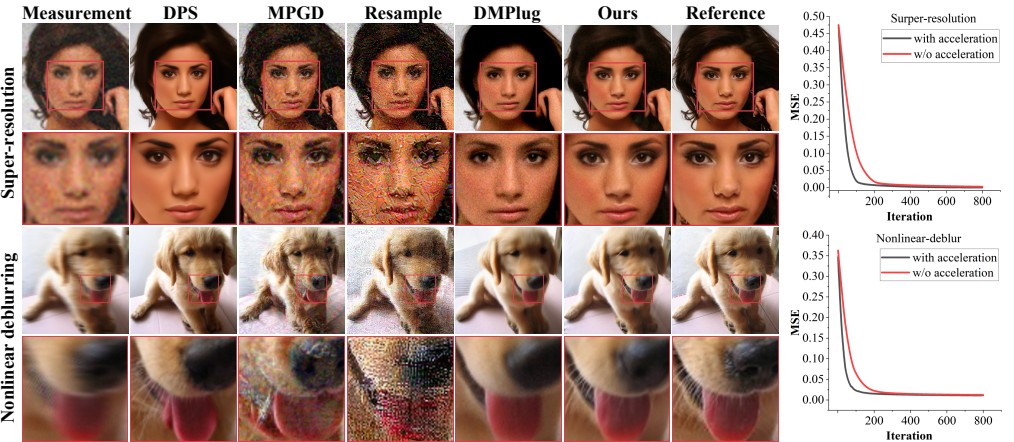

Figure 3: (Left) Sample results of our PDSE and key competing methods for $4\times$ super-resolution and nonlinear deblurring under Gaussian noise ($\sigma = 0.08$). Certain regions are enlarged to facilitate detailed comparison. (Right) Convergence speed with or without our acceleration strategy, i.e., a measurement-consistency gradient correction near $t^*$, for problems of super-resolution and nonlinear deblurring.

Recovery quality is evaluated using PSNR, SSIM, and LPIPS Zhang et al. (2018). The concerned task formulations and more implementation details are given in Appendix C.

## 4.1 LINEAR INVERSE PROBLEMS

**Super-resolution**, **inpainting**, **Gaussian and motion deblur**. Following the procedures outlined in DPS and DMPlug, the image processing tasks are performed using the following settings: *i) $4\times$ bicubic downsampling for super-resolution, ii) inpainting with a random mask covering 70% of pixels, iii) Gaussian blur with a $61 \times 61$ kernel and standard deviation 3, and iv) motion blur with a randomly generated $61 \times 61$ kernel at intensity 0.5.* In these linear settings, All experiments are corrupted by additive noise with a variance 0.01. Quantitative results are reported in Table 1, with qualitative comparisons given in Appendix E. As seen, the proposed PDSE consistently matches or exceeds competing approaches on both datasets. While LPIPS gains are marginal in some cases, our proposal achieves an average PSNR gain of 0.67 dB over the second-best competitors across all tasks.

## 4.2 NONLINEAR INVERSE PROBLEMS

**Nonlinear Deblurring**. For nonlinear deblurring application, the learned blurring operators from Tran et al. (2021) are used, with a known Gaussian-shaped kernel and Gaussian additive noise under a variance of $\sigma = 0.01$, as described in Chung et al. (2022); Song et al. (2024). The results are given in Table 2, which shows that PDSE surpasses the state-of-the-art ReSample, with average gains of 2.016 in PSNR, 0.089 in SSIM, and 0.011 in LPIPS across two datasets.

**BID with Turbulence**. BID seeks to simultaneously recover a sharp image $x$ and an unknown blur kernel $k$ from a degraded observation $y = k * x + n$. For turbulence-induced blur, commonly

| Methods | Dataset | Super-resolution (4×) | | | Non-uniform image deblurring | | |
|---|---|---|---|---|---|---|---|
| | | Guassian | Impulse | Speckle | Guassian | Impulse | Speckle |
| | | Low/High | Low/High | Low/High | Low/High | Low/High | Low/High |
| DPS (ICLR 2023) | CelebA | 26.49/25.53 | 24.83/23.76 | 25.66/24.87 | 24.12/23.48 | 23.59/23.92 | 23.31/23.55 |
| MPGD (ICLR 2024) | | 23.14/21.23 | 18.76/17.05 | 19.01/18.56 | 18.29/18.16 | 17.92/16.54 | 17.03/18.15 |
| ReSample (ICLR 2024) | | 14.56/13.21 | 15.35/13.71 | 15.86/14.35 | 23.25/20.62 | 20.73/18.84 | 23.42/21.78 |
| DMPlug (NeurIPS 2024) | | 27.13/26.23 | 26.41/25.21 | 25.41/24.88 | 27.41/26.53 | 27.35/26.09 | 27.92/26.81 |
| PDSE (Ours) | | **28.54/27.17** | **30.48/30.01** | **26.69/26.18** | **28.15/27.24** | **28.02/26.87** | **28.64/27.35** |
| Ours vs. Best compe. | | ▲1.41/0.94 | ▲4.07/4.80 | ▲1.28/1.30 | ▲0.74/0.71 | ▲0.67/0.78 | ▲0.72/0.54 |
| DPS (ICLR 2023) | ImageNet | 24.87/23.91 | 23.21/22.15 | 24.03/23.24 | 22.58/21.93 | 22.14/22.37 | 21.89/22.01 |
| MPGD (ICLR 2024) | | 21.68/19.87 | 17.42/15.73 | 17.65/17.21 | 17.01/16.89 | 16.58/15.28 | 15.79/16.82 |
| ReSample (ICLR 2024) | | 13.24/12.03 | 14.12/12.58 | 14.53/13.17 | 21.78/19.35 | 19.46/17.61 | 22.01/20.42 |
| DMPlug (NeurIPS 2024) | | 25.68/24.79 | 24.93/23.84 | 24.12/23.61 | 25.89/25.12 | 25.82/24.67 | 26.38/25.35 |
| PDSE (Ours) | | **27.12/25.83** | **28.76/28.32** | **25.34/24.86** | **26.71/25.89** | **26.54/25.48** | **27.19/26.12** |
| Ours vs. Best compe. | | ▲1.44/1.04 | ▲3.83/4.48 | ▲1.22/1.25 | ▲0.82/0.77 | ▲0.72/0.81 | ▲0.81/0.77 |

Table 3: **(Robustness)** Super-resolution and nonlinear deblurring on CelebA and ImageNet under various types and levels of noise. To save space, only the values of PSNR are provided. (**Bold**: Best performance, Underlined: Second best performance, Red: Performance increase).

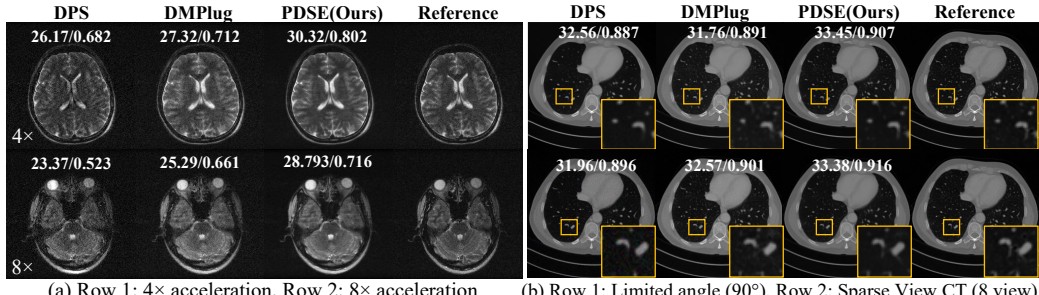

(a) Row 1: 4× acceleration, Row 2: 8× acceleration  (b) Row 1: Limited angle (90°), Row 2: Sparse View CT (8 view)

Figure 4: **Medical imaging**. Representative reconstructions by PDSE and leading baselines. (a) MRI acceleration at $4\times$ and $8\times$. (b) CT reconstruction under limited-angle ($90°$) and sparse-view settings. PSNR and SSIM are presented.

encountered in long-range imaging, the process can be modeled as follows.

$$\boldsymbol{y} = \boldsymbol{k} * T_\theta(\boldsymbol{x}) + \boldsymbol{n},$$

where $T_\theta(\boldsymbol{x})$ represents an unknown tilt transformation. Note BlindDPS employs diffusion models for the image and blur kernel, while DMPlug and PDSE use them specifically for the image. The experimental results are given in Table 2 (b), which shows PDSE again enjoys consistently superior performance, confirming stronger reconstruction capability.

**Experiments with Unknown Noise**. We evaluate robustness to unknown noise on representative linear and nonlinear inverse problems, i.e., super-resolution and non-uniform image deblurring, on CelebA and ImageNet datasets. Measurements incorporate Gaussian, impulse, and speckle noise at low and high intensity levels (Appendix C.1). Table 3 and the left part of Fig. 3 show that ReSample and MPGD struggle with robustness; for example, MPGD suffers a PSNR gap of 5.4–5.9 dB on CelebA and 5.4–6.0 dB on ImageNet compared to our method. PDSE outperforms DMPlug by 4.80 dB on CelebA and 4.48 dB on ImageNet in PSNR for super-resolution under high impulse noise, benefiting from the partial diffusion at $t^*$ and the adaptive stopping mechanism of ALES.

**Experiments with Medical Images**. For experiments on medical images, diffusion models are fine-tuned on $256 \times 256$ resolution MR images randomly sampled from the M4Raw Lyu et al. (2023) training dataset and subsequently tested on 100 distinct MR images from the concerned test set. We adopt the CelebA model from Chung et al. Chung et al. (2023b) as the pre-trained base, performing 10,000 fine-tuning iterations. Coil Sensitivity Maps (CSMs) are estimated using the fastmri library. Since the data comprise real-world samples, no additional noise is further introduced. Experiments are performed at 4-fold and 8-fold acceleration factors. For comparison, two leading algorithms from the natural image domain, DPS and DMplug, are selected as baselines. Fig. 4 (a) presents visual reconstructions highlighting critical anatomical structures. These results visually confirm that our

(a) **(Ablation)** Critical Timestep $t^*$

| Timestep ($t^*$) | 0.1T | 0.2T | 0.3T | 0.4T | 0.5T |
|---|---|---|---|---|---|
| PSNR | 27.30 | 28.80 | **29.54** | 29.20 | 28.90 |
| SSIM | 0.7150 | 0.7550 | **0.7670** | 0.7580 | 0.7450 |
| Timestep ($t^*$) | 0.6T | 0.7T | 0.8T | 0.9T | 1.0T |
| PSNR | 28.50 | 28.10 | 27.70 | 27.10 | 26.80 |
| SSIM | 0.7350 | 0.7200 | 0.7050 | 0.6900 | 0.6800 |

(b) **(Ablation)** Different Learning Rates

| Rate | 0.001 | 0.005 | 0.01 | 0.05 | 0.1 |
|---|---|---|---|---|---|
| PSNR | 29.40 | **29.54** | 28.90 | 28.10 | 26.80 |
| SSIM | 0.7620 | **0.7670** | 0.7450 | 0.7200 | 0.6800 |

(c) **(Efficiency)** Runtime comparison in wall-clock time

| Methods | Wall-clock time [s] | |
|---|---|---|
| | Super-resolution (Linear) | No-uniform deblurring (Nonlinear) |
| DPS (ICLR 2023) | 182.04 | 187.67 |
| MCG (NeurIPS 2022) | 186.71 | 192.21 |
| MPGD (ICLR 2024) | 6.70 | 6.91 |
| DiffPIR (CVPR 2023) | 44.53 | - |
| ReSample (ICLR 2024) | 462.32 | 501.04 |
| DMPlug (NeurIPS 2024) | 1008.18 | 1097.20 |
| PDSE (Ours) | 258.13 | 241.96 |

Table 4: (Left) Ablation on FFHQ super-resolution ($\sigma$=0.01): (a) critical timestep $t^*$, (b) learning rate. (Right) Efficiency on CelebA (RTX A6000 GPU).

method produces smoother images with more accurate and sharper details. More quantitative and visual results are presented in Appendix E.

**Experiments with CT Images**. For CT acceleration, we fine-tuned a CelebA-pretrained model Chung et al. (2023b) for 19,500 iterations on $256 \times 256$ slices from the AAPM Low Dose CT dataset AAPM Grand Challenge Organizers (2016), testing on the L067 subject. The CT imaging process was simulated using the DDS physics module Chung et al. (2023c). Experiments on this clinical data, without added noise, were performed under two regimes: sparse-view with 8 projection angles and limited-angle with 90 projection angles. We compare against DPS and DMPlug baselines. Visual results in Fig. 4 (b) demonstrate our method's ability to recover sharper details, with full quantitative analysis provided in Appendix E.

### 4.3 ABLATION STUDY AND EFFICIENCY ANALYSIS

The main ablation results are given as follows. More details are referred to Appendix E.

**Effectiveness of critical timestep $t^*$**. The role of critical timestep $t^*$ is ablated on the FFHQ dataset for $4\times$ super-resolution under Gaussian noise ($\sigma = 0.01$). Table 4 (a) shows that the best performance is achieved when $t^*$ is around $0.30T$, with PSNR reaching $29.54$ dB, which aligns closely with our theoretical analysis in Section 3.

**Impact of Learning Rate**. The impact of different learning rates on super-resolution performance is ablated on the FFHQ dataset. Table 4 (b) shows that a learning rate of 0.005 yields the best results, indicating that a moderate learning rate better balances convergence and stability in this task.

**Convergence Behavior w/o Acceleration**. We evaluate the convergence of our method in super-resolution and nonlinear deblurring. In the right plot of Fig. 3, the accelerated version (black curve) converges faster than the non-accelerated one (red curve), reducing MSE early and speeding up with fewer iterations, further confirming its role in the efficiency boost.

**Efficiency Analysis**. Table 4 (c) compares runtime efficiency for practices of super-resolution and non-linear deblurring. The proposed PDSE achieves 258.13 seconds for super-resolution and 241.96 seconds for deblurring, outperforming SOTA methods ReSample with 462.32 seconds and 501.04 seconds as well as DMPlug with 1008.18 seconds and 1097.20 seconds.

## 5 CONCLUSION

In this paper, we introduce a partial diffusion model with the aid of score evolution, which leverages the reverse diffusion process starting from a critical score point $t^*$ to boost the efficiency in solving inverse problems. Through theoretical analysis, we identify the optimal balance between signal and noise and demonstrate that our proposal outperforms existing approaches in massive image restoration practices. Experimental results show significant improvements in recovery quality, particularly under noisy conditions. Additionally, the elaboration of an adaptive learning and early stopping scheme accelerates convergence by duly adjusting the early stopping criteria. Overall, PDSE enhances efficiency while retaining the generative strength of the typical diffusion model, fitting better for most practical applications such as image super-resolution, inpainting, and deblurring.

## REPRODUCIBILITY STATEMENT

An implementation of the core algorithm is provided at https://anonymous.4open.science/r/code-of-PDSE. The empirical validation is based on the publicly available datasets (FFHQ, CelebA, M4Raw, AAPM-CT) and pre-trained models cited in this paper. The appendix contains the theoretical proofs for the propositions presented, configurations for each inverse problem, implementation details for all baselines, and a full list of the hyperparameters used to obtain the results.

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

## A  APPENDIX

**The Use of Large Language Models**: A large language model (LLM) was used to help prepare this manuscript. Its main role was to improve the writing, such as correcting grammar and clarifying phrasing. It also helped to double-check a few mathematical formulas and logical steps in the derivations. The LLM was not used for the core research ideas, experiments, or analysis. The core research ideas, methodology, and results presented are the original work of the authors.

## APPENDIX GUIDE

**Sections**

**Tables**

**Figures**

In this section, we provide supplementary results to enrich the findings presented in the main paper. **Section A** presents the theoretical proof of our propositions and elaborates on the implementation of ALES. **Section B** outlines the essential implementation details, including the setup for baseline methods and the configurations of the inverse problems. **Section C** showcases additional experimental results.

## B METHODOLOGY

### B.1 PROOF OF $t^*$ DERIVATION

**Proposition B.1** (Score Transition Threshold). *In variance preserving diffusion models, there exists a unique critical time $t^*$ satisfying $\bar{\alpha}_{t^*} = \frac{1}{2}$ where:*

*1. The signal-to-noise ratio* $\mathrm{SNR}(t^*) = \frac{\mathrm{Var}(\sqrt{\bar{\alpha}_{t^*}}\boldsymbol{x}_0)}{\mathrm{Var}(\sqrt{1-\bar{\alpha}_{t^*}}\epsilon)} = \frac{\bar{\alpha}_{t^*}}{1-\bar{\alpha}_{t^*}} = 1,$

*2. The critical time $t^*$ satisfies the condition $\int_0^{t^*} \beta(s)\,ds = \ln 2$, which determines $t^*$ based on the accumulated variance.*

*Proof.* In typical diffusion models, such as DDPM, the forward process is defined as $x_t = \sqrt{\bar{\alpha}_t}x_0 + \sqrt{1-\bar{\alpha}_t}\epsilon$, where $\epsilon \sim \mathcal{N}(0, I)$ and $\bar{\alpha}_t = \prod_{s=1}^{t}(1-\beta_s)$.

Given $\bar{\alpha}_{t^*} = \frac{1}{2}$, we can compute the signal-to-noise ratio $\text{SNR}(t^*)$ as:

$$\text{SNR}(t^*) = \frac{\bar{\alpha}_{t^*}}{1 - \bar{\alpha}_{t^*}} = 1,$$

which holds true when $\bar{\alpha}_{t^*} = \frac{1}{2}$.

Next, we take the natural logarithm of both sides:

$$\sum_{s=1}^{t^*} \log(1 - \beta_s) = -\ln 2.$$

For small $\beta_s$, we approximate $\log(1 - \beta_s) \approx -\beta_s$, yielding:

$$\sum_{s=1}^{t^*} \beta_s \approx \ln 2.$$

For a continuous DDPM schedule, let $\bar{\alpha}_t = \exp\left(-\int_0^t \beta(s)\, ds\right)$. Then:

$$\exp\left(-\int_0^{t^*} \beta(s)\, ds\right) = \frac{1}{2},$$

which leads to:

$$\int_0^{t^*} \beta(s)\, ds = \ln 2.$$

$\square$

**Example**: In DDPM's discrete linear schedule, $\beta_t = \beta_1 + \frac{t}{T-1}(\beta_T - \beta_1)$, where $t = 0, 1, \ldots, T-1$, forming an arithmetic sequence with first term $\beta_1$, difference $d = \frac{\beta_T - \beta_1}{T-1}$, and $t^* + 1$ terms. Thus:

$$\sum_{s=0}^{t^*} \beta_s = (t^* + 1)\beta_1 + \frac{\beta_T - \beta_1}{T-1} \cdot \frac{t^*(t^* + 1)}{2} = \ln 2.$$

Solving this quadratic in $t^*$, substituting typical DDPM values (e.g., $\beta_1 = 10^{-4}$, $\beta_T = 0.02$, $T = 1000$) yields $t^* \approx 0.26T$. In DDIM, reusing DDPM's schedule also gives $t^* \approx 0.26T$; with reduced $T$ and adjusted $\beta_T$ (e.g., $T = 100$, $\beta_T = 0.1$), the same form shifts $t^*$ to approximately $0.3$–$0.4\, T$.

### B.2 PROOF OF GENERATIVE CAPACITY RETENTION

**Proposition B.2** (Partial Generation Fidelity Bound). *Let $R(\cdot)$ denote the reverse process starting from $t^*$, and let $R_{full}(\cdot)$ be the full reverse process from $t = T$. Under the assumption that $\bar{\alpha}_{t^*} = \frac{1}{2}$, the KL divergence between the generated distribution and the data distribution satisfies:*

$$D_{KL}(p_{data} \| p_{R(x_{t^*})}) \leq \frac{\ln 2}{\int_0^T \beta(s)\, ds} \cdot D_{KL}(p_{data} \| p_{R_{full}(x_T)}). \tag{15}$$

*Proof.* The partial reverse process $R(\cdot) = g_{\epsilon_\theta^{(0)}} \circ \cdots \circ g_{\epsilon_\theta^{(t^*)}}$ starting from $t^*$ where $\bar{\alpha}_{t^*} = \frac{1}{2}$ retains at least a fraction $1 - \frac{\ln 2}{\int_0^T \beta(s)\, ds}$ of the generative capacity of the full process $R_{full}(\cdot) = g_{\epsilon_\theta^{(0)}} \circ \cdots \circ g_{\epsilon_\theta^{(T)}}$, with:

$$1 - \frac{D_{KL}(p_{data} \| p_{R(x_{t^*})})}{D_{KL}(p_{data} \| p_{R_{full}(x_T)})} \geq 1 - \frac{\ln 2}{\int_0^T \beta(s)\, ds}, \tag{16}$$

and computational steps reduced to $\frac{t^*}{T}$. Next, we will outline the specific steps of the proof.

SETUP

In a variance-preserving diffusion model, the forward process is:

$$x_t = \sqrt{\bar{\alpha}_t}x_0 + \sqrt{1 - \bar{\alpha}_t}\epsilon, \quad \epsilon \sim \mathcal{N}(0, I), \tag{17}$$

where $\bar{\alpha}_t = \exp\left(-\int_0^t \beta(s)\,ds\right)$, and $\beta(s)$ is a continuous noise schedule over $t \in [0, T]$. The reverse process iteratively denoises from $x_T$ to $x_0$, with $R_{\text{full}}(x_T)$ running the full trajectory from $x_T$ (typically $\mathcal{N}(0, I)$) to $x_0$, inducing $p_{R_{\text{full}}(x_T)}$, and $R(x_{t^*})$ running from $x_{t^*}$ at $t^*$ to $x_0$, inducing $p_{R(x_{t^*})}$.

GENERATIVE CAPACITY DEFINITION

Generative capacity is measured by the closeness of the generated distribution to $p_{\text{data}}$ via KL divergence. The retention fraction is defined as:

$$1 - \frac{D_{\text{KL}}(p_{\text{data}}\|p_{R(x_{t^*})})}{D_{\text{KL}}(p_{\text{data}}\|p_{R_{\text{full}}(x_T)})}, \tag{18}$$

where $x_{t^*}$ is optimally sampled (e.g., via forward diffusion or conditioning).

NOISE SCHEDULE AND $t^*$

The schedule $\beta(s)$ governs $\bar{\alpha}_t$. At $t^*$, $\bar{\alpha}_{t^*} = \frac{1}{2}$, so:

$$\int_0^{t^*} \beta(s)\,ds = -\ln \bar{\alpha}_{t^*} = \ln 2 \tag{19}$$

For the full process:

$$\bar{\alpha}_T = \exp\left(-\int_0^T \beta(s)\,ds\right), \tag{20}$$

where $\int_0^T \beta(s)\,ds$ is determined by $\beta(s)$ and $T$.

KL DIVERGENCE APPROXIMATION

In the continuous limit, the KL divergence approximates the score estimation error:

$$D_{\text{KL}}(p_{\text{data}}\|p_{R_{\text{full}}(x_T)}) \approx \tag{21}$$

$$\int_0^T \frac{\beta(s)}{2}\mathbb{E}_{p_s(x_s)}[\|\varepsilon_\theta(x_s, s) - \nabla_{x_s}\log p_s(x_s)\|^2]\,ds, \tag{22}$$

and for the partial process:

$$D_{\text{KL}}(p_{\text{data}}\|p_{R(x_{t^*})}) \approx \tag{23}$$

$$\int_0^{t^*} \frac{\beta(s)}{2}\mathbb{E}_{p_s(x_s)}[\|\varepsilon_\theta(x_s, s) - \nabla_{x_s}\log p_s(x_s)\|^2]\,ds. \tag{24}$$

Assuming a well-trained $\varepsilon_\theta$ with approximately constant score error $\mathbb{E}[\|\varepsilon_\theta - \nabla \log p\|^2]$ across $t$:

$$\frac{D_{\text{KL}}(p_{\text{data}}\|p_{R(x_{t^*})})}{D_{\text{KL}}(p_{\text{data}}\|p_{R_{\text{full}}(x_T)})} \approx \frac{\int_0^{t^*} \beta(s)\,ds}{\int_0^T \beta(s)\,ds} = \frac{\ln 2}{\int_0^T \beta(s)\,ds}. \tag{25}$$

Thus, the retention fraction is:

$$1 - \frac{D_{\text{KL}}(p_{\text{data}}\|p_{R(x_{t^*})})}{D_{\text{KL}}(p_{\text{data}}\|p_{R_{\text{full}}(x_T)})} \geq 1 - \frac{\ln 2}{\int_0^T \beta(s)\,ds}. \tag{26}$$

$$\square$$

**Example**: For a linear schedule $\beta(t) = 10^{-4} + \frac{t}{1000}(0.02 - 10^{-4})$, $T = 1000$, $\int_0^T \beta(s)\,ds \approx 10$, and $t^* \approx 0.26T$, the retention is $1 - \frac{\ln 2}{10} \approx 0.93$, with steps reduced by $74\%$.

## B.3 KL BOUND FOR PARTIAL DIFFUSION

**Theorem B.3** (KL Divergence Upper Bound for Truncated Reverse Process). *Assume the reverse process starts at time $t^*$ with $p_{t^*}(\boldsymbol{x}_{t^*}) = q_{t^*}(\boldsymbol{x}_{t^*})$. Let $s^*(\boldsymbol{x}_t, t) = \nabla_{\boldsymbol{x}_t} \log q(\boldsymbol{x}_t)$ be the true score and $s_\theta(\boldsymbol{x}_t, t)$ the learned score. Then,*

$$D_{\mathrm{KL}}(q(\boldsymbol{x}_0)\|p_\theta(\boldsymbol{x}_0)) \leq \frac{1}{2} \int_0^{t^*} \mathbb{E}_{\boldsymbol{x}_t}[\|s_\theta(\boldsymbol{x}_t, t) - s^*(\boldsymbol{x}_t, t)\|^2]\, w(t)\, dt, \tag{27}$$

*where $w(t)$ depends on the variance schedule $\beta(t)$.*

*Proof.* We start by expressing the KL divergence:

$$D_{\mathrm{KL}}(q(\boldsymbol{x}_0)\|p_\theta(\boldsymbol{x}_0)) = \mathbb{E}_q\left[\log \frac{q(\boldsymbol{x}_0)}{p_\theta(\boldsymbol{x}_0)}\right]. \tag{28}$$

Using the diffusion process, we rewrite the KL divergence as:

$$D_{\mathrm{KL}}(q(\boldsymbol{x}_0)\|p_\theta(\boldsymbol{x}_0)) = \mathbb{E}_q\left[\int_0^{t^*} \log \frac{q(\boldsymbol{x}_t)}{p_\theta(\boldsymbol{x}_t)} dt\right]. \tag{29}$$

We now apply score matching, which gives the following bound:

$$\log \frac{q(\boldsymbol{x}_t)}{p_\theta(\boldsymbol{x}_t)} \leq \frac{1}{2} \|s_\theta(\boldsymbol{x}_t, t) - s^*(\boldsymbol{x}_t, t)\|^2. \tag{30}$$

Substituting this bound into the KL divergence expression, we obtain the upper bound:

$$D_{\mathrm{KL}}(q(\boldsymbol{x}_0)\|p_\theta(\boldsymbol{x}_0)) \leq \frac{1}{2}\mathbb{E}_q\left[\int_0^{t^*} \|s_\theta(\boldsymbol{x}_t, t) - s^*(\boldsymbol{x}_t, t)\|^2 w(t) dt\right], \tag{31}$$

where $w(t)$ is a weighting function depending on the variance schedule $\beta(t)$. $\square$

## B.4 ERROR IN TWEEDIE-BASED ONE-STEP RECONSTRUCTION

**Proposition B.4** (Error Bound based on Score Estimation). *The expected squared error of the one-step reconstruction $\hat{x}_0$ obtained from $x_{t^*}$ using the Tweedie formula (based on noise prediction $\epsilon_\theta$) is related to the score estimation error $\delta_s = s_\theta(x_{t^*}, t^*) - s^*(x_{t^*}, t^*)$ as follows:*

$$\mathbb{E}[\|\hat{x}_0 - x_0\|^2] \approx \frac{(1 - \overline{\alpha}_{t^*})^2}{\overline{\alpha}_{t^*}} \mathbb{E}[\|s_\theta(x_{t^*}, t^*) - s^*(x_{t^*}, t^*)\|^2] \tag{32}$$

*Proof.* We start with the standard Tweedie formula for estimating $x_0$ from $x_{t^*}$ using the learned noise prediction network $\epsilon_\theta(x_{t^*}, t^*)$, as given in Eq. (5):

$$\hat{x}_0(x_{t^*}) = \frac{x_{t^*} - \sqrt{1 - \overline{\alpha}_{t^*}}\, \epsilon_\theta(x_{t^*}, t^*)}{\sqrt{\overline{\alpha}_{t^*}}}$$

Recall the forward process $x_{t^*} = \sqrt{\overline{\alpha}_{t^*}} x_0 + \sqrt{1 - \overline{\alpha}_{t^*}} \epsilon$, where $\epsilon \sim \mathcal{N}(0, I)$ is the true noise added. The true data $x_0$ can be expressed as:

$$x_0 = \frac{x_{t^*} - \sqrt{1 - \overline{\alpha}_{t^*}} \epsilon}{\sqrt{\overline{\alpha}_{t^*}}}$$

The reconstruction error is the difference between the estimated $\hat{x}_0$ and the true $x_0$:

$$\hat{x}_0 - x_0 = \frac{(x_{t^*} - \sqrt{1 - \overline{\alpha}_{t^*}} \epsilon_\theta) - (x_{t^*} - \sqrt{1 - \overline{\alpha}_{t^*}} \epsilon)}{\sqrt{\overline{\alpha}_{t^*}}}$$

$$\hat{x}_0 - x_0 = \frac{-\sqrt{1 - \overline{\alpha}_{t^*}} (\epsilon_\theta(x_{t^*}, t^*) - \epsilon)}{\sqrt{\overline{\alpha}_{t^*}}}$$

Let $\delta_\epsilon = \epsilon_\theta(x_{t^*}, t^*) - \epsilon$ denote the error in the noise prediction. Then:

$$\hat{x}_0 - x_0 = -\frac{\sqrt{1 - \overline{\alpha}_{t^*}}}{\sqrt{\overline{\alpha}_{t^*}}} \delta_\epsilon$$

The expected squared reconstruction error is:

$$\mathbb{E}[||\hat{x}_0 - x_0||^2] = \mathbb{E}\left[\left\|-\frac{\sqrt{1 - \overline{\alpha}_{t^*}}}{\sqrt{\overline{\alpha}_{t^*}}} \delta_\epsilon\right\|^2\right] = \frac{1 - \overline{\alpha}_{t^*}}{\overline{\alpha}_{t^*}} \mathbb{E}[||\delta_\epsilon||^2]$$

Now, we relate the noise prediction error $\delta_\epsilon$ to the score estimation error $\delta_s = s_\theta(x_{t^*}, t^*) - s^*(x_{t^*}, t^*)$. The true score function for the VP process is given by:

$$s^*(x_{t^*}, t^*) = \nabla_{x_{t^*}} \log p(x_{t^*}|x_0) = -\frac{x_{t^*} - \sqrt{\overline{\alpha}_{t^*}} x_0}{1 - \overline{\alpha}_{t^*}} = -\frac{\sqrt{1 - \overline{\alpha}_{t^*}} \epsilon}{1 - \overline{\alpha}_{t^*}} = -\frac{\epsilon}{\sqrt{1 - \overline{\alpha}_{t^*}}}$$

Diffusion models are often trained such that the learned noise predictor $\epsilon_\theta$ relates to the learned score $s_\theta$ via $s_\theta(x_{t^*}, t^*) \approx -\frac{\epsilon_\theta(x_{t^*}, t^*)}{\sqrt{1 - \overline{\alpha}_{t^*}}}$. This relationship stems directly from the score matching objective and the definition of the score. Under this approximation, the score error is:

$$\delta_s = s_\theta(x_{t^*}, t^*) - s^*(x_{t^*}, t^*) \approx \left(-\frac{\epsilon_\theta}{\sqrt{1 - \overline{\alpha}_{t^*}}}\right) - \left(-\frac{\epsilon}{\sqrt{1 - \overline{\alpha}_{t^*}}}\right)$$

$$\delta_s \approx -\frac{\epsilon_\theta - \epsilon}{\sqrt{1 - \overline{\alpha}_{t^*}}} = -\frac{\delta_\epsilon}{\sqrt{1 - \overline{\alpha}_{t^*}}}$$

Rearranging this gives the relationship between the noise error and the score error:

$$\delta_\epsilon \approx -\sqrt{1 - \overline{\alpha}_{t^*}} \delta_s$$

Substituting this back into the expression for the expected squared reconstruction error:

$$\mathbb{E}[||\hat{x}_0 - x_0||^2] = \frac{1 - \overline{\alpha}_{t^*}}{\overline{\alpha}_{t^*}} \mathbb{E}[||\delta_\epsilon||^2] \approx \frac{1 - \overline{\alpha}_{t^*}}{\overline{\alpha}_{t^*}} \mathbb{E}[|| - \sqrt{1 - \overline{\alpha}_{t^*}} \delta_s||^2]$$

$$\mathbb{E}[||\hat{x}_0 - x_0||^2] \approx \frac{1 - \overline{\alpha}_{t^*}}{\overline{\alpha}_{t^*}} (1 - \overline{\alpha}_{t^*}) \mathbb{E}[||\delta_s||^2]$$

$$\mathbb{E}[||\hat{x}_0 - x_0||^2] \approx \frac{(1 - \overline{\alpha}_{t^*})^2}{\overline{\alpha}_{t^*}} \mathbb{E}[||s_\theta(x_{t^*}, t^*) - s^*(x_{t^*}, t^*)||^2]$$

This completes the proof, showing the direct relationship between the reconstruction error of the Tweedie estimator and the squared error of the score function approximation. $\square$

## B.5 IMPLEMENTATION OF ALES

The Adaptive Learning Early Stopping (ALES) method, designed to address the "early-learning-then-overfitting" (ELTO) phenomenon in inverse problems, extends prior strategies like ES-WMV Wang et al. (2023) with enhanced robustness. Algorithm 2 outlines its implementation, integrating three key innovations: time-weighted variance, stage-aware adaptation, and local sensitivity detection. Below, we elaborate on each innovation, tying them to the algorithm's structure and their practical benefits.

**1. Time-Weighted Variance (WMV).** Unlike *ES-WMV*'s uniform weighting, ALES uses a linearly time-weighted moving variance to emphasize recent optimization behavior. Given a window size $W$ and image sequence $\{x_t\}$, the WMV at window start $t$ is

$$\text{WMV}(t) = \sum_{w=0}^{W-1} \omega_w \left\|x_{t+w} - \frac{1}{W} \sum_{i=0}^{W-1} x_{t+i}\right\|_F^2, \quad \omega_w = \frac{w+1}{\sum_{j=1}^W j}.$$

Weights $\omega_w$ increase with recency, sharpening responsiveness to abrupt changes while damping stale fluctuations. When fewer than $W$ images are available, WMV is treated as $+\infty$ to avoid premature stops.

---

**Algorithm 2:** ALES: Adaptive Learning and Early Stopping

---

**Input:** window $W$, variance threshold $\delta_v$, loss threshold $\alpha$, patience $P$, minimum iterations
$\qquad E_{\min}$
**Data:** image history $\mathcal{H}$, loss history $\mathcal{L}$, best loss $L_{\text{best}}$, counter $c$
**Output:** stop flag
CheckStopping$(e, x_e, L_e)$
append $x_e$ to $\mathcal{H}$; append $L_e$ to $\mathcal{L}$
**if** $e < E_{\min}$ **then return** False
1.  Time-Weighted Variance
**if** $|\mathcal{H}| \geq W$ **then**
$\quad \mathcal{R} \leftarrow$ last $W$ elements of $\mathcal{H}$; $\mu \leftarrow \frac{1}{W}\sum_{i=1}^{W}\mathcal{R}_i$
$\quad v_{\text{cur}} \leftarrow \sum_{i=1}^{W} \frac{i}{\sum_{j=1}^{W} j} \mathbb{E}\big[(\mathcal{R}_i - \mu)^2\big]$
**else**
$\quad \llcorner\ v_{\text{cur}} \leftarrow +\infty$
2.  Local Sensitivity Analysis
**if** $v_{cur} < \delta_v \wedge e > 1.5 E_{\min} \wedge \text{Var}(\textit{last } W \textit{ in } \mathcal{L}) < 0.01\,\alpha \wedge e > 2E_{\min}$ **then**
$\quad \llcorner$ **return** True
3.  Stage-Aware Adaptation
**if** $L_e < L_{\text{best}}(1-\alpha)$ **then**
$\quad |\ L_{\text{best}} \leftarrow L_e$; $c \leftarrow 0$
**else**
$\quad \llcorner\ c \leftarrow c + 0.5 + \big(0.5 \text{ if } e \geq 1.5 E_{\min} \text{ else } 0\big)$
**if** $c \geq P$ **then**
$\quad \mathcal{S} \leftarrow$ last $P$ in $\mathcal{L}$; $r \leftarrow \frac{1}{P-1}\sum_k |\mathcal{S}_k - \mathcal{S}_{k-1}|$
$\quad$ **if** $r > 2\alpha$ **then** $c \leftarrow P/2$; **return** False
$\quad \llcorner$ **return** True
**return** False

---

**2. Stage-Aware Adaptation.** ALES modulates patience by training phase. When the current loss improves significantly,

$$L_e \;<\; (1-\alpha)\, L_{\text{best}},$$

it updates $L_{\text{best}}$ and resets the counter $c \leftarrow 0$. Otherwise, it increments $c$ more slowly early on and faster later,

$$\Delta c(e) \;=\; 0.5 \;+\; 0.5\,\mathbf{1}[e \geq 1.5\, E_{\min}].$$

Once $c \geq P$, it performs a turbulence check over the last $P$ losses $\mathcal{S}$ by computing

$$r \;=\; \frac{1}{P-1}\sum_k |\mathcal{S}_k - \mathcal{S}_{k-1}|.$$

If $r > 2\alpha$, ALES deems the trajectory unstable, halves the effective patience by setting $c \leftarrow P/2$, and continues training; otherwise it stops. This phased scheme avoids premature halts early and enforces decisive stopping under stable late-stage dynamics, improving upon *ES-WMV*'s fixed-patience design.

**3. Local Sensitivity Detection.** To robustly detect convergence in critical (e.g., edge-rich) regions, ALES couples the image-space WMV with loss-space stability. An early stop is triggered only when

$$\text{WMV} \;<\; \delta_v, \quad \text{Var}\big(\text{recent } W \text{ losses}\big) \;<\; 0.01\,\alpha, \quad e \;>\; 2\,E_{\min}.$$

This multi-criteria gate requires simultaneously low image variance and low loss variability in a sufficiently mature phase, implicitly capturing local sensitivity without explicitly computing $\max\|x_t - \bar{x}\|$. Compared with the purely global criterion in *ES-WMV*, the joint check yields greater robustness in noisy or non-stationary settings.

# C ADDITION IMPLEMENTATION DETAILS

## C.1 MORE DETAILS

**Noise Generation.** To simulate unknown noise in our experiments, we generate three types of noise—Gaussian, impulse, and speckle—each with low and high intensity levels. Details are provided below. Gaussian noise is modeled as zero-mean additive noise with variances of 0.08 and 0.12 for low and high levels, respectively. Impulse noise, often referred to as salt-and-pepper noise, randomly alters each pixel with a probability $p \in [0, 1]$, replacing it with either a white or black pixel (equal probability). We set $p = 0.03$ for low noise and $p = 0.06$ for high noise. Speckle noise is applied to each pixel $x \in [0, 1]$ as $x(1 + \epsilon)$, where $\epsilon$ follows a zero-mean Gaussian distribution with variances of 0.15 and 0.20 for low and high levels, respectively.

**Implementation Details.** We adopt a unified setup across all inverse problems (IPs). The pretrained diffusion models (DMs) used for image generation $R(\cdot)$ are taken from Dhariwal & Nichol (2021)[1] and Choi et al. (2021)[2], with sampling performed via the standard DDIM sampler Song et al. (2021b). The total number of diffusion steps is set to $T = 10$ in all experiments. In tasks involving blur and turbulence, pretrained DMs for the blur kernel and tilt map are adopted from Chung et al. (2023a)[3]. The reconstruction loss $\ell(\cdot)$ is set to the mean squared error (MSE). Optimization is performed using the ADAM optimizer, with the learning rate for the latent variable $z$ set to $5 \times 10^{-3}$. For BID with turbulence, the learning rates for the kernel and tilt map are set to $1 \times 10^{-1}$ and $1 \times 10^{-7}$, respectively. The maximum number of iterations is fixed at 800 for all tasks, which we find sufficient for convergence. All experiments are conducted on NVIDIA A6000 GPUs with 48 GB memory. Besides, both the critical latent state $x_{t*}$ and the step size $\eta$ of the measurement-consistency update are treated as learnable variables and are jointly optimized via Adam during inference.

## C.2 IMPLEMENTATIONS OF BASELINES

**DPS Chung et al. (2023b).** Following the original codebase provided by DPS, we leverage pre-trained models trained on CelebA and FFHQ datasets to address both linear and nonlinear tasks. The task-specific hyperparameters are configured following the default settings in the code for consistency and optimal performance.

**MCG Chung et al. (2022).** The implementation of MCG is based on the DPS codebase. Originally designed for inpainting, MCG defines the projection term as $(I - M)X + MY$, where $M$ is a mask. By extending $M$ to a general degradation operator $A$, the projection term is reformulated as $X - A^\top(AX - Y)$. Consequently, the projection term in DPS is modified from $(I - A^\top A)Y - AX$ to $X - A^\top(AX - Y)$, while all other parameters remain consistent with those in the original DPS.

**DMPlug Wang et al. (2024).** Building upon the DMPlug codebase, the implementation adopts the version without early stopping while leveraging the same pre-trained models as DPS to ensure consistency. The diffusion process employs 3 DDIM steps, with optimization performed using the Adam optimizer at a learning rate of 0.02. To achieve a balance between computational efficiency and robust performance, the training process spans 2000 epochs.

**ReSample Song et al. (2024).** Following the ReSample codebase, we use pre-trained LDM-VQ4 models Rombach et al. (2022) trained on FFHQ and CelebA-HQ, with autoencoders generating $64 \times 64 \times 3$ images. All parameters strictly adhere to the original resample configurations.

**MPGD He et al. (2024).** The implementation follows the original code from the MPGD repository, utilizing the FFHQ diffusion model provided by DPS and the VQGAN from the latent diffusion model by Rombach et al. (2022). Default parameters from MPGD are adopted to maintain consistency with the original setup.

---

[1]https://github.com/openai/guided-diffusion
[2]https://github.com/jychoi118/ilvr_adm?tab=readme-ov-file
[3]https://github.com/BlindDPS/blind-dps

**DiffPIR Zhu et al. (2023).**    We adopt the official implementation of DiffPIR and employ the same pre-trained diffusion model as used in our paper. The measurement model and degradation settings are kept consistent with our proposed method to ensure a fair comparison. All hyperparameters are tuned to achieve the best possible performance under each task.

**BlindDPS Chung et al. (2023a).**    The official implementation of BlindDPS is used, including its BID turbulence-related modules. All experimental settings and hyperparameters strictly follow those provided in the original paper to ensure optimal performance.

The used default code and settings of each competitor's official implementation, listed below.

• DPS Chung et al. (2023b): `https://github.com/DPS2022/diffusion-posterior-sampling`

• MCG Chung et al. (2022): `https://github.com/hyungjin-chung/MCG_diffusion`

• MPGD He et al. (2024): `https://github.com/KellyYutongHe/mpgd_pytorch`

• DMPlug Wang et al. (2024): `https://github.com/soominkwon/resample/tree/main`

• ReSample Song et al. (2024): `https://github.com/soominkwon/resample/tree/main`

• BlindDPS Chung et al. (2023a): `https://github.com/sun-umn/DMPlug`

• DiffPIR Zhu et al. (2023): `https://github.com/yuanzhi-zhu/DiffPIR`

## C.3 CONFIGURATION DETAILS OF INVERSE PROBLEM SETTINGS

For super-resolution, inpainting, gaussian deblurring, motion deblurring, nonlinear deblurring and BID with turbulence, we use the forward models from the of main competing methods Chung et al. (2022); He et al. (2024);

**Super-resolution.**    Noisy image super-resolution aims to reconstruct a clean RGB image $x$ from its noisy, down-sampled observation $y = \mathcal{D}(x) + n$. The downsampling operator, $\mathcal{D}(\cdot)$ : $[0,1]^{3 \times tH \times tW} \rightarrow [0,1]^{3 \times H \times W}$, reduces the input image dimensions $tH \times tW$ by a factor of $t$, while $n$ represents additive noise. In our experiments, we set $t = 4$. To ensure fair comparisons, the formulation excludes any explicit regularization terms. The reconstruction problem is formulated as:

$$x_{t^*}^* = \arg \min_{x_{t^*}} \ell\big(y, \mathcal{D}(\mathcal{R}(x_{t^*}))\big),$$

**Inpainting.**    Noisy image inpainting involves reconstructing a clean RGB image $x \in [0,1]^{3 \times H \times W}$ that is partially observed and further contaminated by additive noise $n$. The forward model is described as $y = m \odot x + n$, where $m \in \{0,1\}^{3 \times H \times W}$ is a binary mask, and $\odot$ denotes the Hadamard product. Given $y$ and $m$, the goal is to recover $x$. Consistent with Song et al. (2024), the masks for all three channels of $m$ are identical, with 70% of the mask values randomly set to 0. To ensure fair comparisons, no explicit regularization terms are included in the formulation. The reconstruction problem is expressed as:

$$x_{t^*}^* = \arg \min_{z} \ell\big(y, m \odot \mathcal{R}(x_{t^*})\big),$$

**Nonlinear deblurring.**    The setup for nonlinear deblurring follows Chung et al. (2022), which builds upon the approach introduced by Tran et al. (2021). Recently, Tran et al. (2021) proposed learning data-driven blurring models using paired blurry-sharp training sets of the form $\{(y_i, x_i)\}_{i=1,\dots,N}$. The training objective is defined as:

$$\alpha^*, \beta^* = \arg \min_{\alpha,\beta} \sum_{i=1}^{N} \|y_i - \mathcal{F}_\alpha(x_i, \mathcal{G}_\beta(x_i, y_i))\|,$$

where $\mathcal{G}_\beta(\cdot, \cdot)$ predicts the latent blur kernel associated with the input blurry-sharp image pair, and $\psi_\beta(\cdot, \cdot)$ models real-world nonlinear blurring processes based on the input image-kernel pair.

To evaluate the performance of the DPS method on nonlinear inverse problems (IPs), Chung et al. (2022) introduces a specific nonlinear deblurring problem with a known Gaussian-shaped blur kernel:

$$y = \mathcal{F}_{\alpha^*}(x, g) + n,$$

where $g \in \mathbb{R}^{64 \times 64}$ is Gaussian-shaped with $\sigma = 3.0$. The goal is to reconstruct $x$ from $y$ using the forward model $\mathcal{F}_{\alpha^*}(\cdot, g)$. Our formulation adheres to Chung et al. (2022) and does not include additional regularization terms. The reconstruction process is expressed as:

$$x_{t^*}^* = \arg\min_{x_{t^*}} \ell\big(y, \mathcal{F}_{\alpha^*}(\mathcal{R}(x_{t^*}), g)\big),$$

**Gaussian deblurring.** Gaussian blur aims to smooth an image $x \in [0, 1]^{3 \times H \times W}$ by convolving it with a Gaussian kernel $g$. The forward model is expressed as:

$$y = \mathcal{F}_g(x) + n,$$

where $g \in \mathbb{R}^{k \times k}$ is a Gaussian kernel with kernel size $k$ and standard deviation $\sigma$, $\mathcal{F}_g(\cdot)$ denotes the convolution operation, and $n$ represents additive noise. In this setup, the kernel size is set to $k = 61$, and the intensity, represented by the standard deviation, is $\sigma = 3.0$. Gaussian blur is symmetric, which simplifies the transpose operation to an identity mapping. The reconstruction problem is then formulated as:

$$\arg\min_{x_{t^*}} \ell\big(y, \mathcal{F}_g(\mathcal{R}(x_{t^*}))\big),$$

**Motion deblurring.** Motion blur models the directional blur caused by movement during image capture. The forward model is expressed as:

$$y = \mathcal{F}_m(x) + n,$$

where $\mathcal{F}_m(\cdot)$ denotes convolution with a motion blur kernel $m \in \mathbb{R}^{k \times k}$, $k$ is the kernel size, and $n$ represents additive noise. Motion blur differs from Gaussian blur due to its anisotropic effects, as it introduces streaks defined by the direction and intensity of motion. The kernel size is set to $k = 61$, and the intensity parameter, representing the magnitude of the blur, is set to $\sigma = 0.5$. These parameters dictate the extent and orientation of the motion blur effect. The reconstruction task is then formulated as:

$$\arg\min_{x_{t^*}} \ell\big(y, \mathcal{F}_m(\mathcal{R}(x_{t^*}))\big),$$

**BID with Turbulence.** This variant of blind image deconvolution (BID) arises in long-range imaging degraded by atmospheric turbulence and is modeled as a *tilt-then-blur* process Chan (2022): $y = k * T_\phi(x) + n$, where $T_\phi(\cdot)$ applies a spatially varying displacement field $\phi$ such that each pixel $p_i$ is shifted to $p_i + \phi_i$. The goal is to jointly estimate the clean image $x$, blur kernel $k$, and tilt field $\phi$ from the measurement $y$, making this task more ill-posed than standard BID. Following BlindDPS Chung et al. (2023a), we simulate blur using a Gaussian PSF ($\sigma = 3.0$) and generate tilt fields as i.i.d. Gaussian noise over the spatial grid. While BlindDPS employs pretrained diffusion models (DMs) for all components, we use pretrained DMs only for the image. The kernel is constrained to the probability simplex via SoftMax, and the tilt field is initialized with low-variance Gaussian noise and optimized with a small learning rate. Our optimization objective is:

$$\arg\min_{x_{t^*}, k, \phi} \ell\left(y, \text{SoftMax}(k) * T_\phi(R(x_{t^*}))\right) \tag{33}$$

## D ADDITIONAL DISCUSSION

Our exploration of Partial Diffusion with Score Evolution (PDSE) has demonstrated its considerable strengths in tackling diverse inverse problems. We offer here some further reflections on specific aspects of the framework, considering avenues for continued research and refinement.

**Further Considerations for the Critical Timestep $t^*$.** The theoretically-grounded selection of the critical timestep $t^*$, typically where $\overline{\alpha}_{t^*} \approx \frac{1}{2}$ as detailed in Section 3 and Appendix B.1, underpins PDSE's efficiency by guiding its optimal *starting noise level*. Practically, the number of steps in our partial reverse diffusion, $N_R$, must be kept modest. This is because significant memory

costs associated with backpropagation through $R(\cdot)$ preclude employing an $N_R$ that might naively correspond in scale to $t^*$ when $t^*$ is interpreted as a potentially large segment of a full diffusion process. Our ablations, presented in Table 4 (a) where the $t^*/T$ term effectively varies $N_R$ relative to a common reference scale, therefore explore optimal choices for such a constrained, short $N_R$. Specifically, when the total timestep $T = 10$, the empirically identified optimum of $N_R \approx 3$ represents the best trade-off between reconstruction fidelity and these practical memory limitations when initiating the reverse process near the theoretically-indicated noise regime. This choice of a compact $N_R$ is thus a pragmatic adaptation driven by computational resource limits, rather than a conceptual discrepancy with the theory that guides the starting noise level. Of course, the optimal short $N_R$ for a given $t^*$ noise level can subtly shift with different noise schedules or underlying model architectures.

**Adaptability and Tuning of the ALES Mechanism**. The Adaptive Learning and Early Stopping (ALES) strategy, detailed in Algorithm 2 and Appendix B.5, is crucial for PDSE's robust performance against the ELTO phenomenon. This robustness is especially vital under unknown noise conditions where simpler stopping criteria often prove insufficient. ALES achieves this through a principled multi-component design. Key elements like time-weighted variance and stage-aware adaptation specifically address the complexities of optimization within such noisy and varied environments. Although this design involves several parameters, our findings show ALES operates effectively across diverse experiments using consistent default hyperparameters. This demonstrates its practical utility without requiring laborious, task-specific tuning. Such reliable out-of-the-box performance with default settings establishes a strong foundation.

**Broader Utility of Pre-trained Models with PDSE**. PDSE is similar to other leading approaches Chung et al. (2023b); Wang et al. (2024) and builds on rich priors learned by high quality pretrained diffusion models. This reliance reflects current state of the art practice. Future work should evaluate PDSE when base models are trained on constrained or highly specialized datasets common in scientific domains. Parameter efficient fine tuning and domain adaptation for adapting large models to PDSE may broaden its applicability.

**More Complex Nonlinearities**. The robust performance of PDSE on the nonlinear inverse problems tested, such as nonlinear deblurring and BID with turbulence, is encouraging. These tasks highlight the capability of the optimization framework to effectively incorporate the partial diffusion prior. The universe of nonlinear problems is vast, however, and includes scenarios where the forward operator $\mathcal{A}$ might be exceptionally complex or only partially known. Extending the PDSE paradigm to tackle such frontiers, perhaps by integrating it with techniques for implicit operator learning or more structured iterative schemes, remains a compelling research direction.

**Generalizability to New Data and Tasks**. We have validated PDSE across a spectrum of inverse problems and image types, including natural and medical imagery. The underlying principle of partial diffusion, rooted in the score evolution of the diffusion process, suggests a fundamental applicability. A valuable next step would be to empirically explore PDSE's extension to other data modalities, such as 3D volumetric data, video sequences, or even signals in other domains like audio processing. Such studies would further map the landscape of its generalizability and practical utility.

## E  MORE RESULTS

We provide supplementary results for PDSE, including a small set of additional ablations and complementary quantitative/qualitative studies.

**Impact of the Number of Iterative Optimization Steps $N$ without ALES** Table 5 shows that increasing the fixed iteration count improves PSNR and SSIM up to about $N = 400$, after which the gains stall and can decline due to early learning followed by overfitting. With ALES, PDSE reaches $29.54\,\text{dB}$ PSNR and surpasses the best fixed $N$ while removing the need to tune $N$. ALES automatically selects an effective stopping point, limits overfitting, and delivers robust high-quality reconstructions.

**Impact of the Extrapolation Strategy** Table 6 reports an ablation of the extrapolation strategy on CelebA (100-image average). With extrapolation, PDSE achieves slightly better PSNR/SSIM and lower LPIPS while reducing the required iterations from roughly $350$ to about $200$. This shows

| Fixed Iterations $N$ | PSNR (dB) ↑ | SSIM ↑ | LPIPS ↓ |
|---|---|---|---|
| $N = 100$ | 28.30 | 0.7350 | 0.095 |
| $N = 200$ | 28.90 | 0.7480 | 0.083 |
| $N = 300$ | 29.25 | 0.7570 | 0.074 |
| $N = 400$ | 29.40 | 0.7610 | 0.068 |
| $N = 600$ | 27.85 | 0.7150 | 0.112 |
| $N = 800$ | 26.52 | 0.6830 | 0.145 |
| PDSE (with ALES) | **29.54** | **0.7670** | **0.062** |

Table 5: Performance Comparison for Different Fixed Iteration Counts $N$. (FFHQ, $4\times$ Super-Resolution, $\sigma = 0.01$)

| Settings | PSNR (dB) ↑ | SSIM ↑ | LPIPS ↓ | Convergence (Iter) |
|---|---|---|---|---|
| w/o Extrapolation | 29.41 | 0.764 | 0.064 | $\sim$350 |
| **w/ Extrapolation (Ours)** | **29.54** | **0.767** | **0.062** | $\sim$**200** |

Table 6: Ablation of the extrapolation strategy on CelebA (100-image average).

that the extrapolation step primarily accelerates convergence without sacrificing final reconstruction quality.

**Quantitative Analysis of MRI Reconstruction** Table 7 summarizes results on M4Raw. PDSE attains the best PSNR and SSIM at $4\times$ and $8\times$, outperforming DPS and DMPlug. At $4\times$ it reaches 30.97 dB PSNR and 0.804 SSIM; at $8\times$ it achieves 28.81 dB and 0.725. DMPlug yields lower LPIPS, yet PDSE's advantages in PSNR and SSIM indicate more faithful anatomical reconstruction. Aggregate improvements appear in the row *Ours vs. Best compe.*.

| Methods | M4Raw ($4\times$) | | | M4Raw ($8\times$) | | |
|---|---|---|---|---|---|---|
| | PSNR↑ (Mean±Std) | SSIM↑ (Mean±Std) | LPIPS↓ (Mean±Std) | PSNR↑ (Mean±Std) | SSIM↑ (Mean±Std) | LPIPS↓ (Mean±Std) |
| DPS (ICLR 2023) | $25.525 \pm 1.016$ | $0.674 \pm 0.058$ | $0.132 \pm 0.013$ | $22.927 \pm 1.041$ | $0.514 \pm 0.065$ | $0.252 \pm 0.072$ |
| DMPlug (NeurIPS 2024) | $27.017 \pm 0.319$ | $0.707 \pm 0.012$ | $0.125 \pm 0.048$ | $25.202 \pm 0.366$ | $0.652 \pm 0.018$ | $0.154 \pm 0.022$ |
| PDSE (Ours) | $\mathbf{30.966 \pm 0.670}$ | $\mathbf{0.804 \pm 0.014}$ | $\mathbf{0.071 \pm 0.011}$ | $\mathbf{28.806 \pm 0.480}$ | $\mathbf{0.725 \pm 0.015}$ | $\mathbf{0.121 \pm 0.015}$ |
| Ours vs. Best compe. | ▲ 3.949 | ▲ 0.097 | ▼ 0.054 | ▲ 3.604 | ▲ 0.073 | ▼ 0.033 |

Table 7: **(MRI reconstruction)** Quantitative results of MRI reconstruction on the M4Raw dataset under $4\times$ and $8\times$ acceleration rates. (**Bold**: Best performance, Underlined: Second best performance, Red: Performance improvement.)

**Hyperparameter Sensitivity of ALES** Table 8 reports a sensitivity analysis of ALES on CelebA $4\times$ super-resolution (20-image average) by varying the window size $W$, patience $P$, and relative loss threshold $\alpha$ around the default setting. Across all tested configurations, PSNR and SSIM remain essentially unchanged while the average iteration count stays in a stable range of roughly 300–400 steps, indicating that ALES is robust to moderate hyperparameter changes and works well with a single default configuration.

**Effect of Partial Diffusion Steps and Comparison with DMPlug** Table 9 compares PDSE and DMPlug under different numbers of partial diffusion steps on a subset of 20 CelebA images with total diffusion steps $T = 10$. PDSE consistently attains higher PSNR and SSIM than DMPlug at the same number of partial steps, while peak memory decreases as the number of steps is reduced. Notably, PDSE with only two partial steps already outperforms DMPlug with three steps and uses comparable or less memory, and a one-step DMPlug configuration fails to implement (N/A).

**Iteration and NFE Comparison Across Methods** Table 10 summarizes the diffusion steps, external iterations, and total NFEs for representative baselines and PDSE. Sampling-based methods such as DPS and MCG rely on long diffusion chains without backpropagation, while optimization-based methods like DMPlug and PDSE use only a few diffusion steps per update but require many external iterations. With ALES, PDSE typically converges in about 300–400 iterations, corresponding to

.

| Parameter | Variation | PSNR (dB) ↑ | SSIM ↑ | Avg. Iterations |
|---|---|---|---|---|
| Default | $W = 10, P = 20, \alpha = 10^{-3}$ | 29.54 | 0.767 | 342 |
| Window Size $W$ | $W = 5$ | 29.51 | 0.766 | 338 |
| | $W = 8$ | 29.53 | 0.767 | 341 |
| | $W = 12$ | 29.54 | 0.767 | 343 |
| | $W = 15$ | 29.55 | 0.767 | 345 |
| Patience $P$ | $P = 15$ | 29.48 | 0.765 | 315 |
| | $P = 25$ | 29.51 | 0.766 | 365 |
| | $P = 30$ | 29.46 | 0.764 | 393 |
| Threshold $\alpha$ | $5 \times 10^{-4}$ | 29.57 | 0.768 | 368 |
| | $8 \times 10^{-4}$ | 29.55 | 0.767 | 353 |
| | $1.5 \times 10^{-3}$ | 29.48 | 0.765 | 331 |
| | $2 \times 10^{-3}$ | 29.42 | 0.764 | 326 |

Table 8: **(ALES sensitivity)** Hyperparameter sensitivity of ALES on CelebA 4× super-resolution (20-image average).

| Method | Partial Steps | PSNR (dB) ↑ | SSIM ↑ | Peak Memory (MB) |
|---|---|---|---|---|
| PDSE | 3 | 31.20 | 0.8986 | 13,283 |
| PDSE | 2 | 30.53 | 0.8848 | 9,611 |
| PDSE | 1 | 28.54 | 0.8182 | 5,573 |
| DMPlug | 3 | 29.14 | 0.7490 | 12,737 |
| DMPlug | 2 | 25.72 | 0.5890 | 9,063 |
| DMPlug | 1 | N/A | N/A | N/A |

Table 9: **(Diffusion steps)** Comparison of partial diffusion steps between PDSE and DMPlug on CelebA (20-image subset, $T = 10$).

roughly 900–1200 NFEs, which is competitive with or better than DPS while avoiding the need to tune a fixed iteration budget.

| Method | Type | Diff. steps / update | External iters | Total computation |
|---|---|---|---|---|
| DPS | Sampling | 1000 | 1 | 1k NFEs |
| MCG | Sampling | 1000 | 1 | 1k NFEs |
| MPGD | Sampling | 50 | 1 | 50 NFEs |
| ReSample | Sampling | 665 | ~2.5k | 665 NFEs+ 2500 calls |
| DiffPIR | Sampling | 100 | 1 | 100 NFEs |
| DMPlug | Optimization | 3 | 2k–3k | 6k–9k NFEs |
| PDSE (Ours) | Optimization | 3 | 300–400 | 0.9k–1.2k NFEs |

Table 10: **(Iterations and NFEs)** Comparison of diffusion steps, iterations, and NFEs across methods.

**Effect of Measurement Noise Level** Table 11 reports PDSE performance on CelebA (20 test images) for 4× super-resolution and Gaussian deblurring under noisy ($\sigma = 0.01$) and noise-free settings. Removing measurement noise consistently yields higher PSNR/SSIM and lower LPIPS, confirming that PDSE benefits from more accurate forward measurements; in the main paper we adopt the noisy setting as a more challenging and realistic benchmark.

**High-Resolution Experiments on CelebA-512** Table 12 reports PDSE performance on CelebA-512 (20 test images) for 4× super-resolution and Gaussian deblurring using a 256px diffusion model with a patch-based reconstruction strategy. Each 512×512 image is divided into overlapping 256×256 patches, processed independently by PDSE, and then merged by averaging in the overlapping regions. PDSE maintains competitive quality at higher resolutions for these spatially local operators,

| Task | Noise level $\sigma$ | PSNR (dB) ↑ | SSIM ↑ | LPIPS ↓ |
|------|---------|-------------|--------|---------|
| Super-resolution (4×) | 0.01 (noisy) | 30.82 | 0.776 | 0.078 |
| | 0 (noise-free) | 31.24 | 0.789 | 0.072 |
| Gaussian deblurring | 0.01 (noisy) | 30.15 | 0.758 | 0.091 |
| | 0 (noise-free) | 30.68 | 0.767 | 0.085 |

Table 11: **(Noise level)** PDSE performance under noisy and noise-free measurements on CelebA (20-image subset).

supporting the scalability of the partial-diffusion design to larger images when the forward operator is local.

| Task | PSNR (dB) ↑ | SSIM ↑ | LPIPS ↓ |
|------|-------------|--------|---------|
| Super-resolution (512×512, 4×) | 30.18 | 0.795 | 0.053 |
| Gaussian deblurring (512×512) | 30.05 | 0.784 | 0.115 |

Table 12: **(High resolution)** PDSE results on CelebA-512 (20-image subset) for 4× super-resolution and Gaussian deblurring.

**Behavior under Extreme Degradations and Failure Cases** Table 13 summarizes PDSE performance on CelebA under increasingly challenging inpainting and Gaussian deblurring settings by varying the missing ratio and noise level. PDSE maintains reasonable PSNR/SSIM and low LPIPS under moderate degradations (e.g., up to about 90% random missing pixels or Gaussian noise scale $\sigma \leq 0.4$), while performance gradually deteriorates when measurements become extremely sparse (98% missing) or noise-dominated. In these extreme cases, the diffusion prior can still yield visually plausible images but may introduce color-block artifacts and localized texture distortions in facial regions, indicating that the forward information is no longer sufficient to fully constrain the reconstruction.

| Scenario | PSNR (dB) ↑ | SSIM ↑ | LPIPS ↓ |
|----------|-------------|--------|---------|
| Inpainting (80% random missing, $\sigma = 0.01$) | 32.03 | 0.9140 | 0.062 |
| Inpainting (90% random missing, $\sigma = 0.01$) | 29.17 | 0.8791 | 0.072 |
| Inpainting (98% random missing, $\sigma = 0.01$) | 19.46 | 0.6334 | 0.199 |
| Gaussian deblur ($\sigma = 0.2$) | 26.96 | 0.7360 | 0.165 |
| Gaussian deblur ($\sigma = 0.4$) | 25.79 | 0.7010 | 0.187 |
| Gaussian deblur ($\sigma = 0.5$) | 23.05 | 0.6190 | 0.210 |

Table 13: **(Failure cases)** PDSE behavior under extreme degradations on CelebA.

| Methods | Limited-angle 90° | | Sparse-view (8 views) | |
|---------|-------------------|---|----------------------|---|
| | PSNR↑ (Mean±Std) | SSIM↑ (Mean±Std) | PSNR↑ (Mean±Std) | SSIM↑ (Mean±Std) |
| DPS (ICLR 2023) | $31.81 \pm 0.96$ | $0.896 \pm 0.008$ | $33.61 \pm 2.22$ | $0.907 \pm 0.022$ |
| DMPlug (NeurIPS 2024) | $33.27 \pm 1.65$ | $0.907 \pm 0.154$ | $35.63 \pm 1.54$ | $0.915 \pm 0.062$ |
| PDSE (Ours) | $\mathbf{35.39 \pm 0.79}$ | $\mathbf{0.923 \pm 0.007}$ | $\mathbf{37.29 \pm 1.51}$ | $\mathbf{0.936 \pm 0.011}$ |
| Ours vs. Best compe. | ▲ 2.12 | ▲ 0.016 | ▲ 1.66 | ▲ 0.021 |

Table 14: **(CT reconstruction)** Quantitative results under limited-angle 90° and sparse-view (8 views). (**Bold**: Best performance, Underlined: Second best performance, Red: Performance improvement.)

**Quantitative Analysis of CT Reconstruction** Table 14 shows that PDSE attains the best PSNR and SSIM in both limited-angle 90° and sparse-view 8-view settings, outperforming DPS and DMPlug. The gains are consistent across cases and persist under variability, indicating robust reconstruction quality. Summary improvements are given in the "Ours vs. Best compe." row.

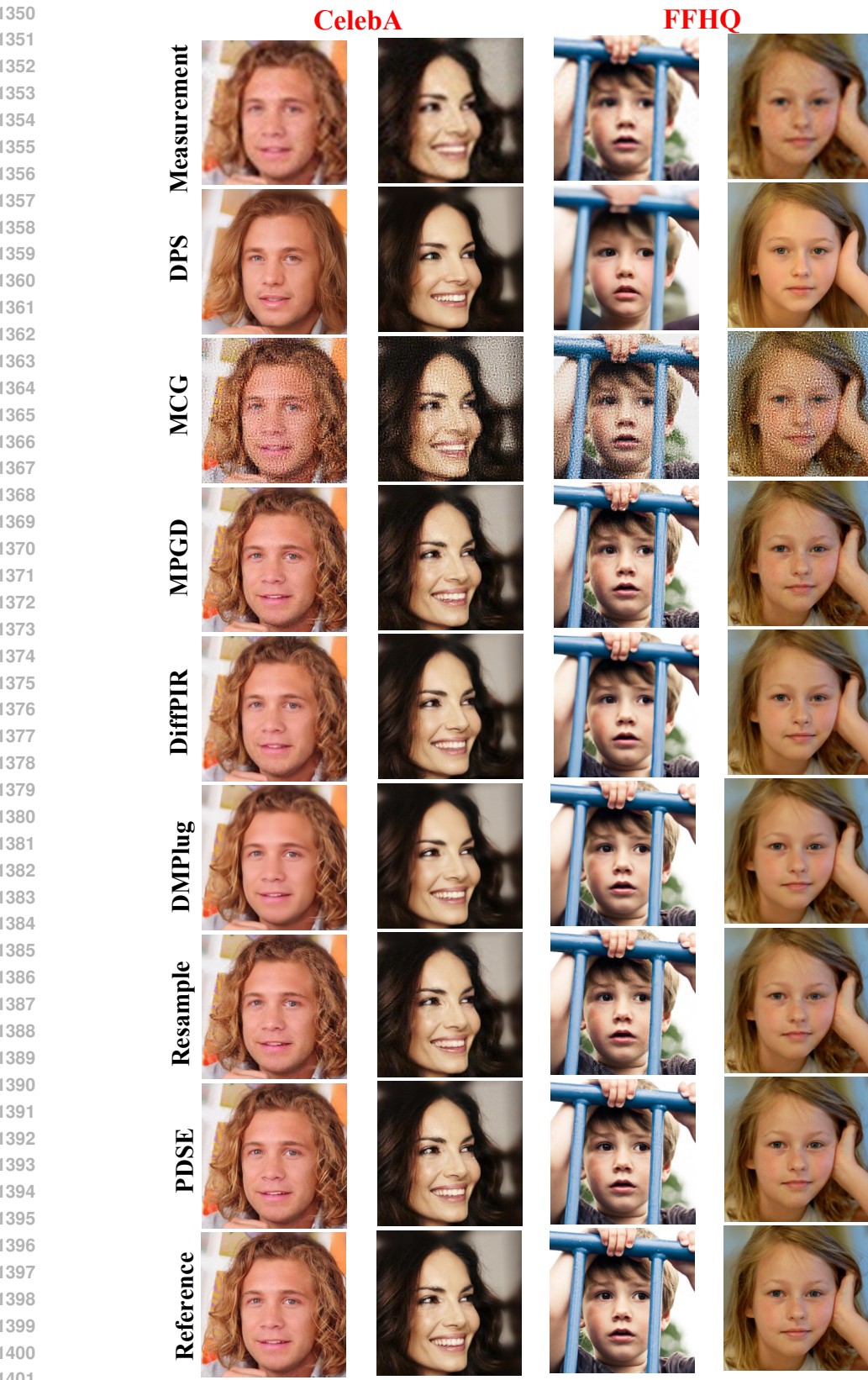

Figure 5: **(Linear IP)** Visualization of representative results produced by the proposed PDSE and competing approaches under 4× super-resolution. All measurements are corrupted with Gaussian noise at a standard deviation of $\sigma = 0.01$.

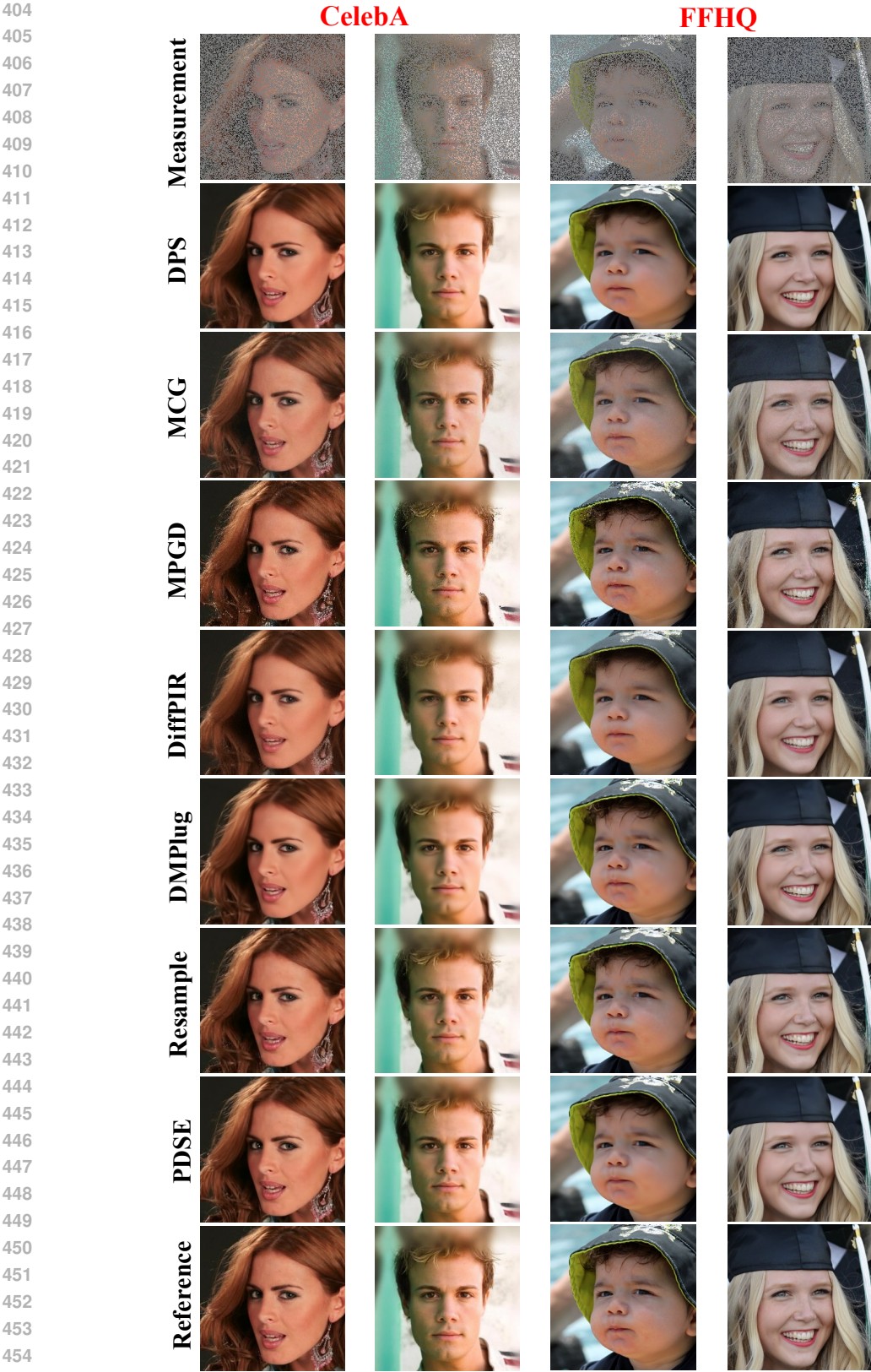

Figure 6: **(Linear IP)** Visualization of representative inpainting results (random 70% missing) obtained by the proposed PDSE and competing approaches. All measurements are corrupted with Gaussian noise with a standard deviation of $\sigma = 0.01$.

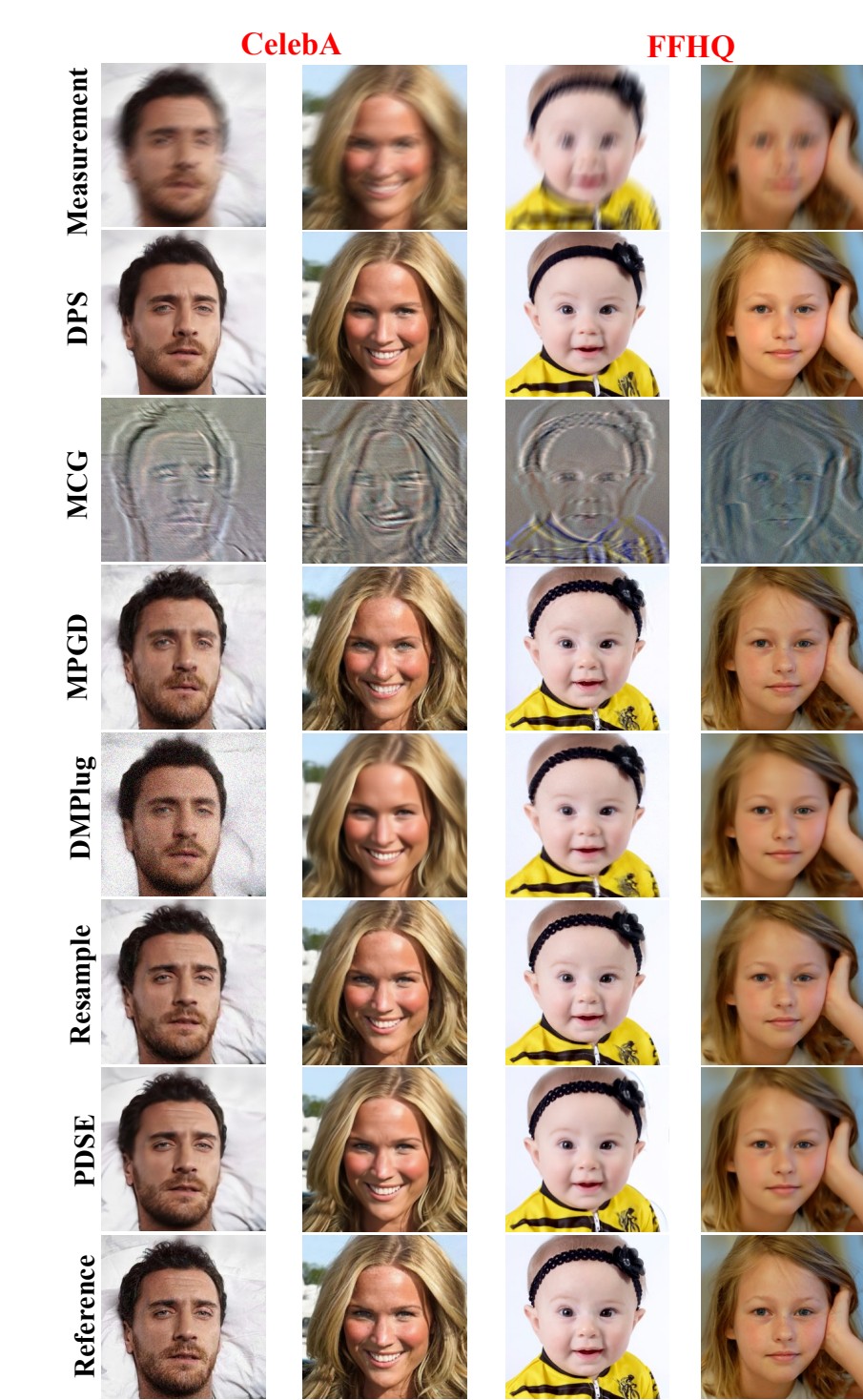

Figure 7: **(Linear IP)** Visualization of representative results produced by the proposed PDSE and competing approaches under motion deblurring. All measurements are corrupted with Gaussian noise at a standard deviation of $\sigma = 0.01$.

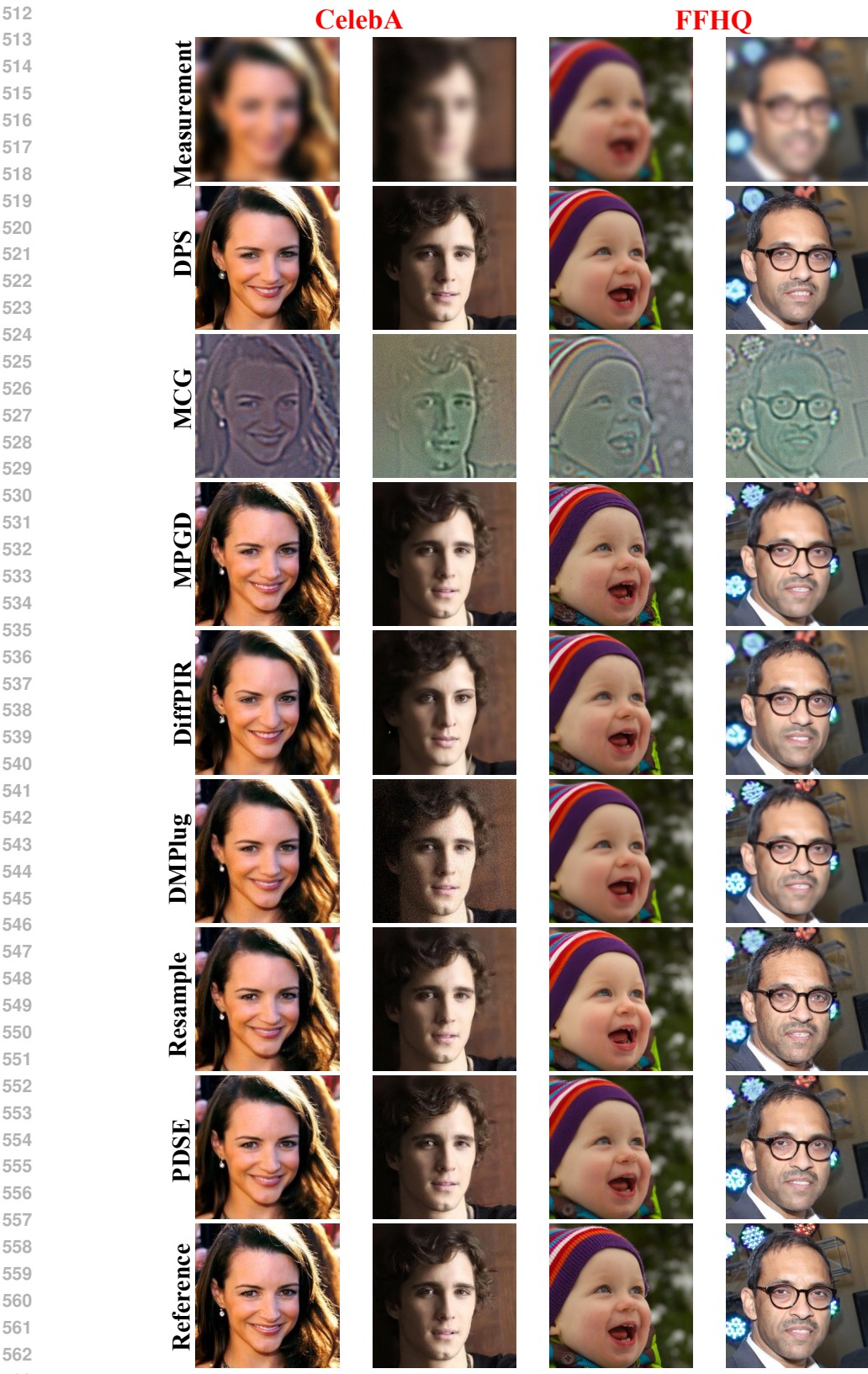

Figure 8: **(Linear IP)** Visualization of representative results produced by the proposed PDSE and competing approaches under gaussian deblurring. All measurements are corrupted with Gaussian noise at a standard deviation of $\sigma = 0.01$.

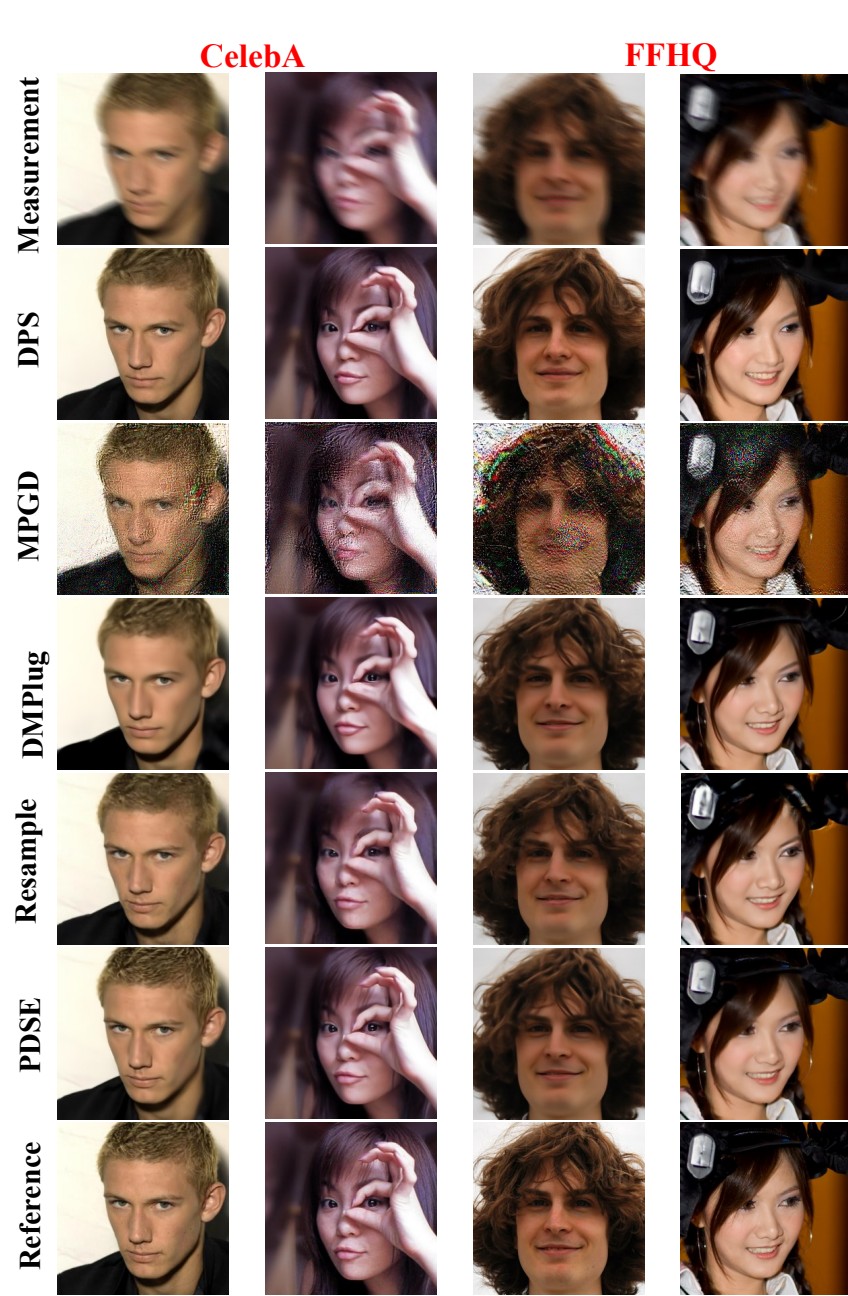

Figure 9: **(Nonlinear IP)** Visualization of representative results produced by the proposed PDSE and competing approaches under nonlinear deblurring. All measurements are corrupted with Gaussian noise at a standard deviation of $\sigma = 0.01$.

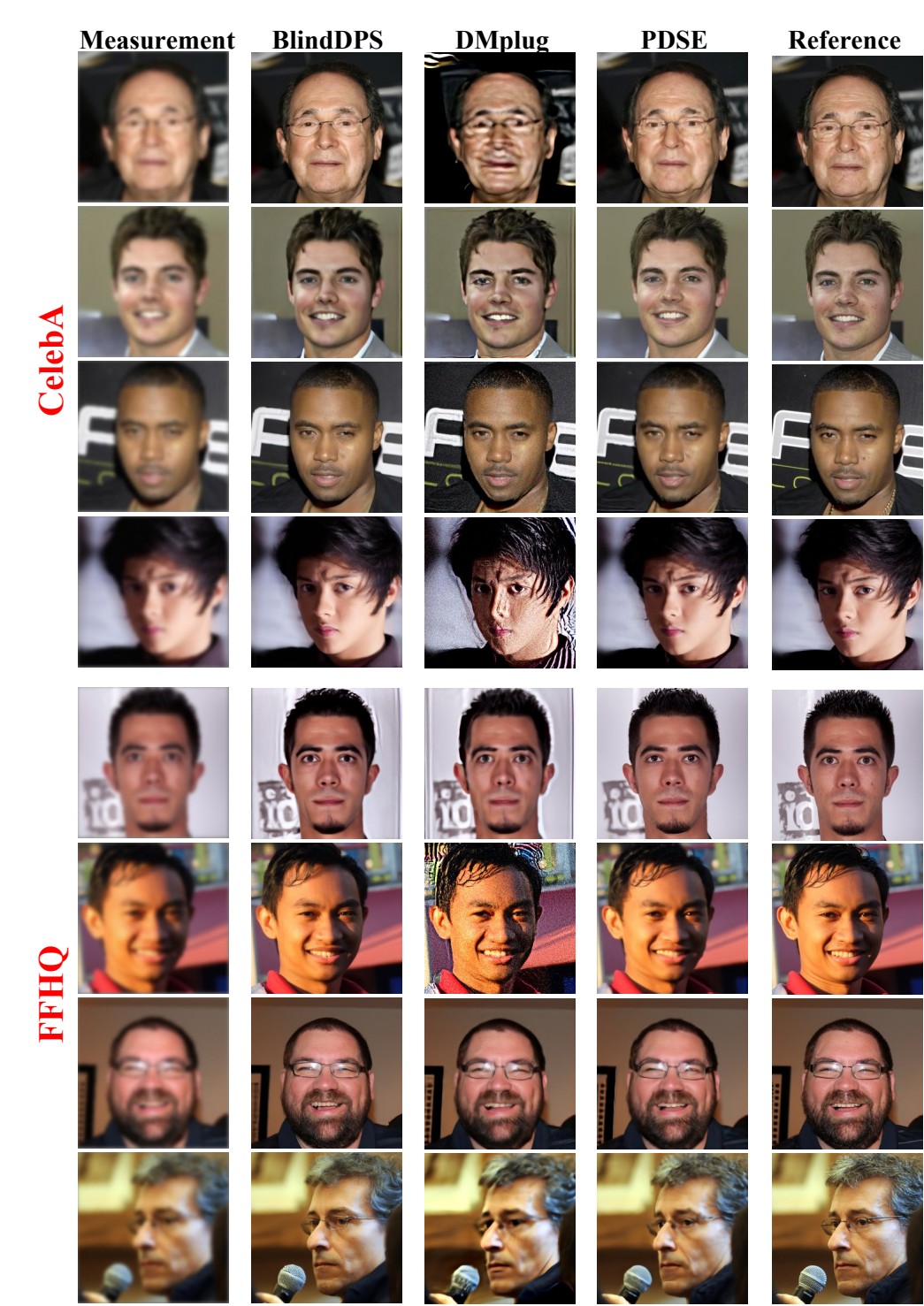

Figure 10: **(Nonlinear IP)** Visualization of representative results produced by the proposed PDSE and competing approaches under BID turbulence. All measurements are corrupted with Gaussian noise at a standard deviation of $\sigma = 0.01$.

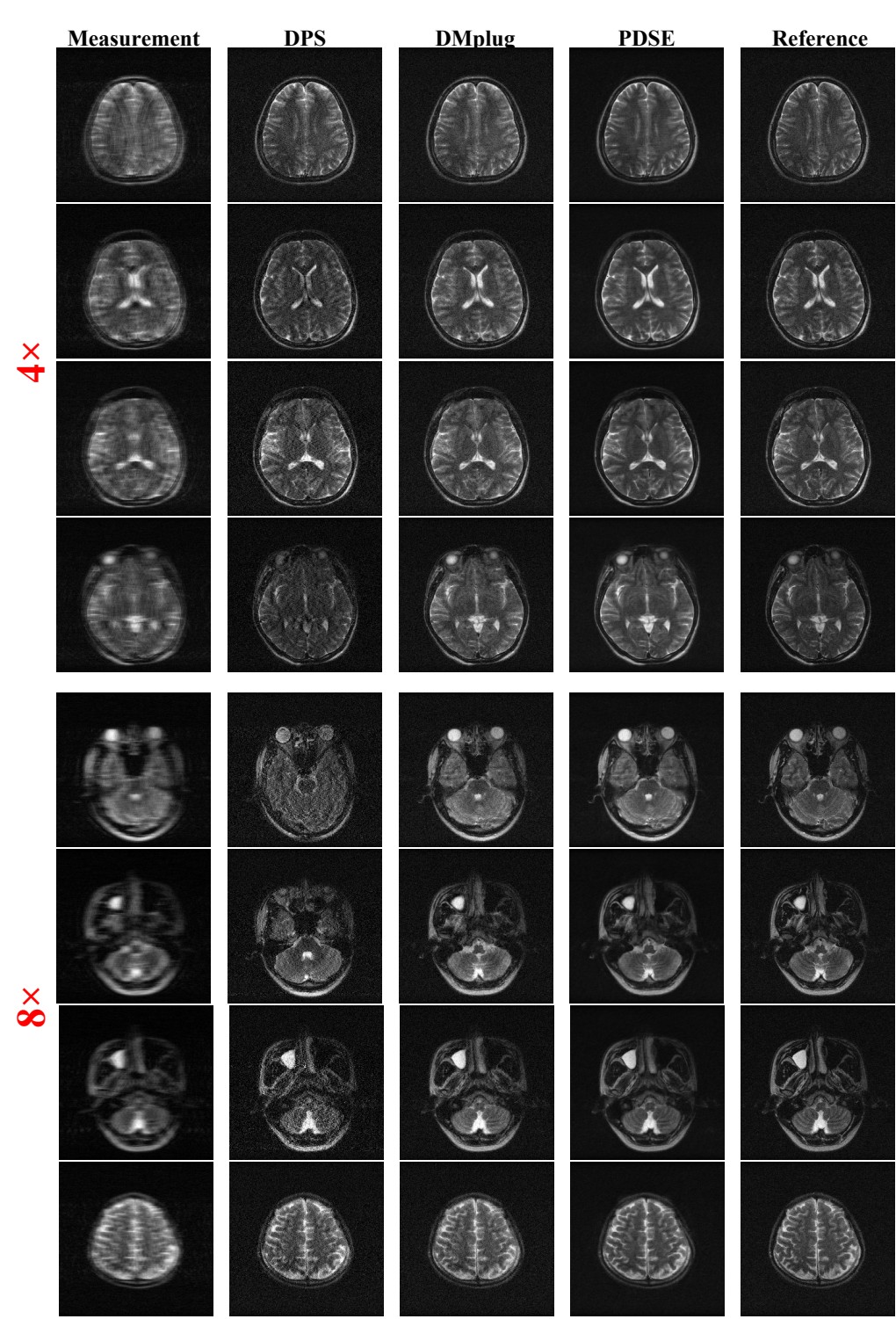

Figure 11: **(MRI reconstruction)** Visualization of representative results produced by the proposed PDSE and competing methods for MRI reconstruction. All measurements are generated following the `fastMRI` protocol, with no additional noise introduced.

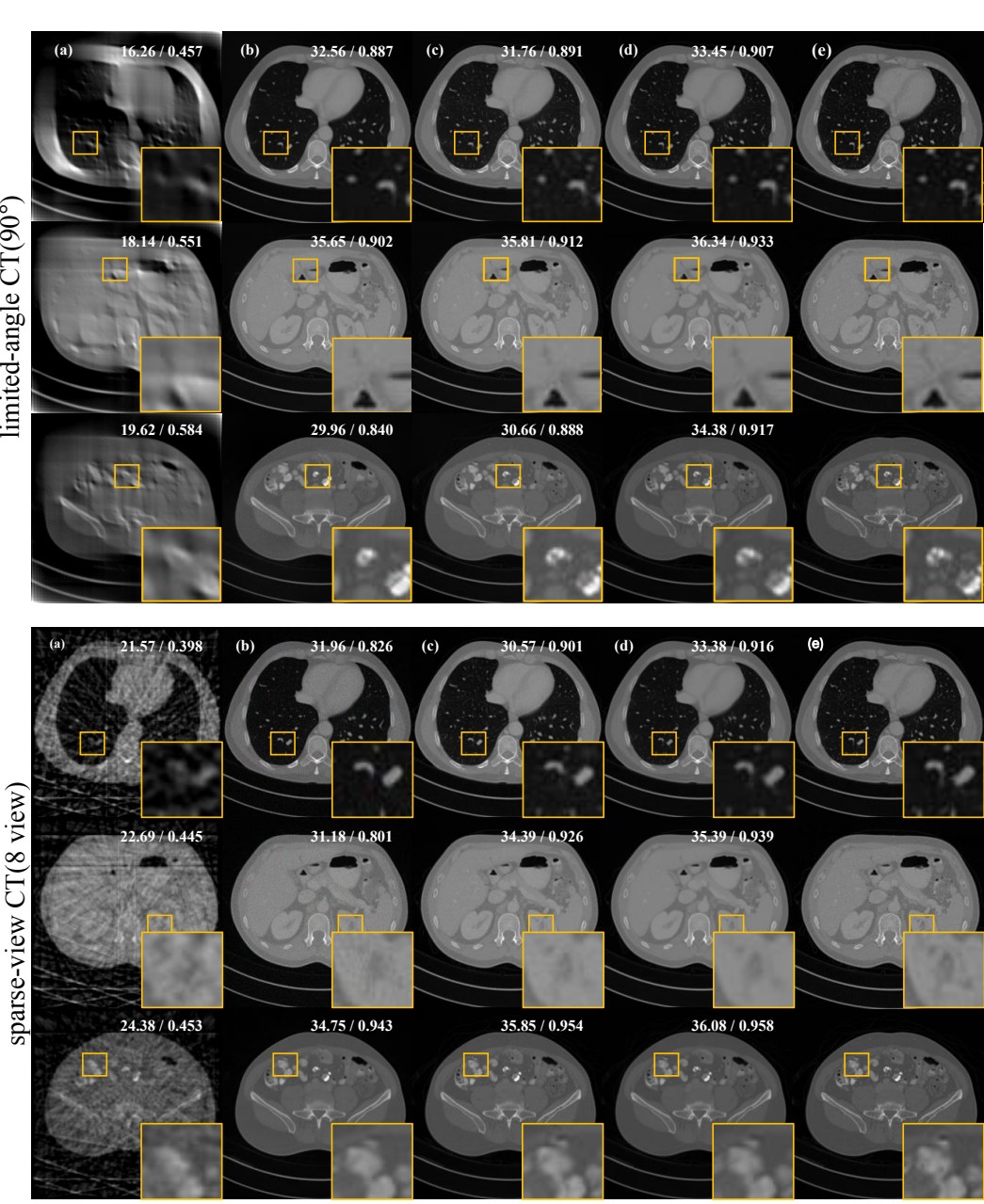

Figure 12: Comparison of CT reconstruction results under different configurations. **(Top): Sparse-view CT**, **(Bottom): Limited-angle CT**. Methods: (a) FBP, (b) DPS, (c) DMPlug, (d) Ours. Numbers in top-right corners denote PSNR and SSIM, respectively.

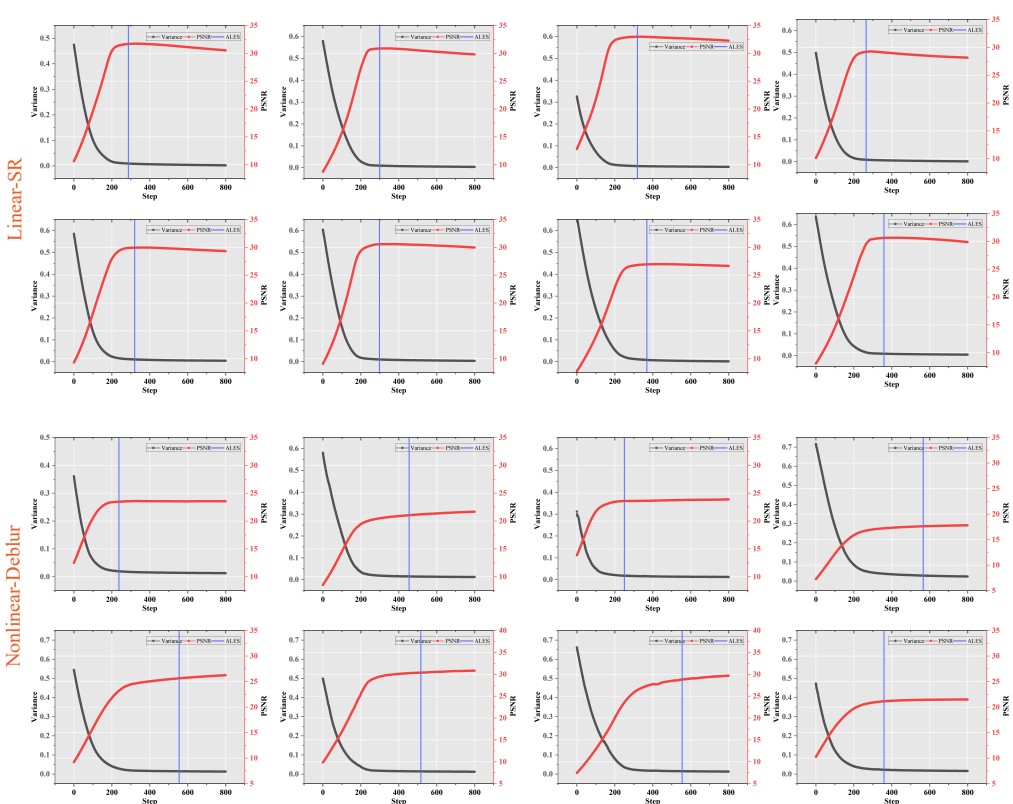

Figure 13: **(ALES)** PSNR and variance evolution of our PDSE method with ALES for linear $4\times$ super-resolution and nonlinear deblurring. Blue curves represent PSNR trajectories, gray curves denote variance trends, and blue bars indicate the detected early stopping points.

