# OpenReview forum: "Partial Diffusion Suffices: Solving General Inverse Problems via Score Evolution"
_ICLR.cc/2026/Conference — ICLR 2026 Conference Desk Rejected Submission_

### Official Review · Reviewer_JPGo · 2025-10-18

**Soundness:** 2
**Presentation:** 1
**Contribution:** 2
**Rating:** 2
**Confidence:** 3

**Summary:**

This paper introduces Partial Diffusion with Score Evolution (PDSE), a new framework for solving inverse problems using conditional score-based generative models (CSGMs). The core idea is to improve the efficiency and quality of CSGM-based approaches, which typically optimize a latent variable (e.g., initial noise $x_T$) by backpropagating through the entire diffusion chain. The author proposed a novel initialization technique that uses partially diffused samples $x_t$ at $t$. Along with the proposed early stopping technique, the proposed method improves CSGM's efficiency and quality.

**Strengths:**

1. The paper identifies a key bottleneck in existing CSGM-style methods: the high computational and memory cost of backpropagating through the full diffusion.
2. The core idea of starting from an intermediate timestamp t* is simple and effective. It naturally reduces the number of sampling steps required for optimization.
3. The authors provide a clear theoretical derivation for their choice of t* based on the SNR=1 equilibrium point.
4. The method demonstrates state-of-the-art or competitive performance against several recent baselines.

**Weaknesses:**

1. The introduction claims to address "two significant gaps": slow convergence and "exploding memory". While the partial diffusion strategy clearly addresses convergence, the solution for memory costs (the ALES-based early stopping) is not a core part of the PDSE formulation itself and is only detailed in the appendix. This makes the writing inconsistent.

2. In section 3.1, the novelty of initializing from a partially-noised state is questionable. This concept is well-explored in related literature (DiffPure [1], SDEdit [2]), and these highly relevant works are not cited. Proposition 3.1 can be considered as a special case of their analysis. A recent work, FMPlug [3], seems to have a more generalized form of the initialization, which allows data-adaptive timestamp optimization.

3. In section 3.2, the explanation of extrapolation is confusing. Taking eq. 13 into eq. 14, the extrapolation step is $\hat{x}^k_\text{out} = \hat{x}^k + \lambda \mu \nabla_{\hat{x}^k} \ell$, which makes the $\lambda$ not very useful. While the illustration of the gradient flow is poorly illustrated, the Alg 1 seems to also contradict it.

4. Section 3.3’s focus is on the issue of previous work. The author should at least provide the improved formulation in the main body and explain why ALES can help with memory issues.

5. The entire experimental results of image restoration task in the main paper (Tables 1 & 2, Figure 3) are conducted exclusively on human face datasets (FFHQ and CelebA). To support the paper's claim of solving "general inverse problems", a more diverse set of results should be presented and analyzed in the main text.

6. The paper lacks crucial ablations in the main text.

$\quad$ a) The effect of the "extrapolation" strategy is only shown in a convergence plot, not with a quantitative table (PSNR/SSIM/LPIPS) to show its impact on final image quality.

$\quad$ b) The hyperparameters for the extrapolation strategy are not discussed or ablated.



[1] Nie, Weili, et al. "Diffusion models for adversarial purification." arXiv preprint arXiv:2205.07460 (2022).

[2] Meng, Chenlin, et al. "Sdedit: Guided image synthesis and editing with stochastic differential equations." arXiv preprint arXiv:2108.01073 (2021).

[3] Wan, Yuxiang, et al. "FMPlug: Plug-In Foundation Flow-Matching Priors for Inverse Problems." arXiv preprint arXiv:2508.00721 (2025).

**Questions:**

1. The optimal timestamp t* is derived from the prior's SNR=1 condition and appears to be fixed (e.g., 0.26T ) regardless of the task. Can you provide empirical analysis of how the type and severity of the degradation (e.g., 90% inpainting mask vs. 10%, or heavy blur vs. light blur) affect the true optimal t*? Is a single, fixed t* really optimal for all inverse problems?

2. In Section 3.2, the author mentions a gradient being "backpropagated through the entire computational graph, including the extrapolation and the decoder R(·)". Can the author clarify this gradient flow and provide the specific hyperparameters used for this acceleration (e.g., the stepsize $\mu$ and scaler$ \lambda$)?

3. The acceleration strategy shows faster convergence in Figure 3 (right). How does this compare, in both reconstruction quality (PSNR/SSIM) and wall-clock time, to the baseline method (w/o acceleration) simply run with a correspondingly larger learning rate? Is this a principled acceleration or an approximation of a more aggressive descent?

4. While the paper claims to mitigate "exploding memory", the total number of diffusion steps $T=10$ which can cause more memory issues compared with DMPlug’s default setting $T=3$. Here are the questions for the author:

$\quad$ a) How does the ALES with a larger number of diffusion steps help with the memory issue?

$\quad$ b) How does the peak GPU memory usage of PDSE compare to a standard CSGM (e.g. DMPlug)?

$\quad$ c) How does the total number of diffusion steps affect the proposed methods’ performance?

---

> ### Author Response · Authors · 2025-11-20
> **Thanks for your valuable review**
>
> ### Your opinions have been of great help to our manuscript. Below, we reply to your questions point-to-point.
> > The introduction claims to address "two significant gaps": slow convergence and "exploding memory". While the partial diffusion strategy clearly addresses convergence, the solution for memory costs (the ALES-based early stopping) is not a core part of the PDSE formulation itself and is only detailed in the appendix. This makes the writing inconsistent.
>
> **Response:**
>
> Thank you for raising this point; it made us realize that the sentence in the introduction describing ALES is not as clear as it should be. **Our core method, partial diffusion (PDSE), is what addresses both slow convergence and memory usage, while ALES is an auxiliary robustness tool for unknown or varying degradations.** The main paper is organized around PDSE, which addresses both gaps by starting from an intermediate timestep $t^*$ and backpropagating only through this short segment, thereby reducing the sampling steps, the depth of the computational graph, and thus peak memory usage. ALES, in contrast, is an adaptive early-stopping strategy for settings with unknown or varying degradations, where the appropriate number of iterations cannot be fixed in advance, so its details are placed in the appendix. In the revision, we will reword this ALES sentence to make this division explicit: PDSE is responsible for both gaps, and ALES is presented as an additional robustness tool rather than the main solution to memory costs.
>
> > In section 3.1, the novelty of initializing from a partially-noised state is questionable. This concept is well-explored in related literature (DiffPure [1], SDEdit [2]), and these highly relevant works are not cited. Proposition 3.1 can be considered as a special case of their analysis. A recent work, FMPlug [3], seems to have a more generalized form of the initialization, which allows data-adaptive timestamp optimization.
>
> **Response:**
>
> We appreciate this comment and agree that initializing from a partially noised state has been explored in prior work. **Our contribution is a different route tailored to visual inverse problems: PDSE turns the partially noised state into an optimization variable tied to a measurement-consistency loss, instead of using it only as a sampling heuristic.** SDEdit [2] and DiffPure [1] use the intermediate state only to shorten denoising, and FMPlug [3] uses foundation flow-matching priors with data-adaptive timestamps, without defining such an optimized intermediate variable or analyzing diffusion score/SNR evolution.  As detailed in Sec. 3, PDSE yields a theoretically guided, optimization-based CSGM that directly targets inverse-problem objectives and works with standard diffusion priors under diverse degradations. We will add and discuss SDEdit, DiffPure, and FMPlug in the revised related work section.
>
>
> > In section 3.2, the explanation of extrapolation is confusing. Taking eq. 13 into eq. 14, the extrapolation step is , which makes the  not very useful. While the illustration of the gradient flow is poorly illustrated, the Alg 1 seems to also contradict it.
>
> **Response:**
> Thank you for highlighting this point. We originally decomposed the update into a “correction” term and a “mixing” term to emphasize their roles, but they together define one measurement-consistency gradient update in the image space. Following your suggestion, the revised paper now presents this operation directly as
> $\hat{x} _ {\text{out}} ^ k = \hat{x} ^ k - \eta \nabla _ {\hat{x} ^ k} \ell(y, \mathcal{A}(\hat{x} ^ k))$. This is a notational simplification: the underlying update we implement remains the same, and the conclusions and performance reported in the experiments are unaffected. To keep the presentation consistent with this single-step form, we have made small clarifying edits to the gradient-flow figure, and Algorithm 1 in the revised PDF.
>
> > Section 3.3’s focus is on the issue of previous work. The author should at least provide the improved formulation in the main body and explain why ALES can help with memory issues.
>
> **Response:**
>
> Thanks. As clarified, the main mechanism that reduces memory cost is PDSE’s partial‑diffusion design, and ALES is used as a robust early‑stopping rule for cases with unknown noise levels. Given the strict page limit, we keep the exposition in the main body focused on PDSE and provide a summary of ALES in Section 3.3, with the complete formulation and implementation placed in the appendix as a practical tool for handling unknown noise.

---

> ### Author Response · Authors · 2025-11-20
> **part 2**
>
> > The entire experimental results of image restoration task in the main paper (Tables 1 & 2, Figure 3) are conducted exclusively on human face datasets (FFHQ and CelebA). To support the paper's claim of solving "general inverse problems", a more diverse set of results should be presented and analyzed in the main text.
>
> **Response:**
>
> Thanks for the concern about dataset diversity. Our initial effort was to test the generality of PDSE across **different image types and forward operators**, so the main paper already includes results on **medical MRI and CT reconstruction** in addition to FFHQ/CelebA restoration. Following your suggestion, it is necessary to add more experiments on general natural images, so we have added experiments on ImageNet, the quantitative results are shown in Table R1, and visual results would be added in the revised PDF. These results support that PDSE is not restricted to faces and can handle diverse image domains and degradations.
>
> **Table R1: Comparison of reconstruction quality across methods on ImageNet (averaged over 100 randomly sampled images) for super-resolution and nonlinear deblurring.**
>
> | Methods | Super-Resolution (4×) (PSNR / SSIM / LPIPS) | Nonlinear Deblurring (PSNR / SSIM / LPIPS) |
> | :---    | :---:                                       | :---:                                      |
> | **DPS** (ICLR'23)      | 27.13 / 0.636 / 0.145 | 26.78 / 0.687 / 0.144 |
> | **ReSample** (ICLR'24) | 30.63 / 0.813 / 0.115 | 29.88 / 0.799 / 0.143 |
> | **DMPlug** (NeurIPS'24)| 31.53 / 0.857 / 0.110 | 28.58 / 0.780 / 0.153 |
> | **PDSE (Ours)**        | **32.03** / **0.884** / **0.107** | **31.28** / **0.862** / **0.125** |
>
> > The paper lacks crucial ablations in the main text.
> > a) The effect of the "extrapolation" strategy is only shown in a convergence plot, not with a quantitative table (PSNR/SSIM/LPIPS) to show its impact on final image quality.
>
> **Response:**
>
>  Thanks. In the main paper, the right of Fig. 3 illustrates that the acceleration step tends to speed up convergence while having limited effect on the final reconstruction quality. To make this clearer, Table R2 below reports averages over 100 CelebA test images, indicating that the extrapolation strategy changes PSNR/SSIM/LPIPS only slightly but reduces the required iterations.
>
>
> **Table R2: Ablation of the extrapolation strategy on CelebA (100-image average).**
>
> | Settings                          | PSNR | SSIM | LPIPS | Convergence (Iter) |
> | :-------------------------------- | :--: | :--: | :---: | :----------------: |
> | w/o Extrapolation                 | 29.41 | 0.764 | 0.064 | ~350               |
> | **w/ Extrapolation (Ours)**       | **29.54** | **0.767** | **0.062** | **~200** |
>
> > b) The hyperparameters for the extrapolation strategy are not discussed or ablated.
>
> **Response:**
>
> Thank you for the suggestion. In the revised formulation, the extrapolation strategy is expressed as a single measurement‑consistency gradient step with stepsize $\eta$, so there is only one effective hyperparameter. In practice, we initialize $\eta$ to 0.5 and treat it as a learnable parameter that is optimized jointly during inference.
>
>
> > The optimal timestamp t* is derived from the prior's SNR=1 condition and appears to be fixed (e.g., 0.26T) regardless of the task. Can you provide empirical analysis of how the type and severity of the degradation (e.g., 90% inpainting mask vs. 10%, or heavy blur vs. light blur) affect the true optimal t*? Is a single, fixed t* really optimal for all inverse problems?
>
> **Response:**
>
> Thank you for this important question. The choice of $t ^ *$ is theoretically motivated by the SNR=1 condition, which balances signal and noise and provides a task-agnostic starting point. In practice, reconstruction quality depends on multiple factors including the prior's match to the data, forward operator accuracy, and degradation severity. Our empirical experience shows that $t ^ *$ can vary within $0.2T$ to $0.4T$ with similar results, indicating robustness rather than strict optimality. We set $t ^ * \approx 0.3T$ (to obtain an integer number of diffusion steps) as a principled default that reduces backpropagation depth while maintaining strong fidelity across diverse tasks. For extreme degradations, the benefit of adaptive $t ^ *$ selection may be limited.

---

> ### Author Response · Authors · 2025-11-20
> **part 3**
>
> > In Section 3.2, the author mentions a gradient being "backpropagated through the entire computational graph, including the extrapolation and the decoder R(·)". Can the author clarify this gradient flow and provide the specific hyperparameters used for this acceleration (e.g., the stepsize and scaler)?
>
> **Response:**
>
> Thank you for pointing out the need to clarify the gradient flow and hyperparameters of the acceleration step. In our current formulation (Sec. 3.2 and Algorithm 1), each iteration treats $x _ {t ^ *}$ as a trainable variable: we decode it via the partial reverse process to obtain $\hat{x} ^ k = R(x _ {t ^ *})$, apply a single measurement-consistency gradient step $\hat{x} _ {\text{out}} ^ k = \hat{x} ^ k - \eta \nabla _ {\hat{x} ^ k} \ell(y, \mathcal{A}(\hat{x} ^ k))$, and define the loss on $\hat{x} _ {\text{out}} ^ k$, which is then backpropagated through this update and $R(\cdot)$ so that both $x _ {t ^ *}$ and the effective strength of the acceleration step are learned jointly via Adam.
>
> > The acceleration strategy shows faster convergence in Figure 3 (right). How does this compare, in both reconstruction quality (PSNR/SSIM) and wall-clock time, to the baseline method (w/o acceleration) simply run with a correspondingly larger learning rate? Is this a principled acceleration or an approximation of a more aggressive descent?
>
> **Response:**
>
> Thanks. Without acceleration, we compute the loss on $\hat{x} = R(x _ {t ^ *})$ and backpropagate to $x _ {t ^ *}$. With acceleration, we apply an additional measurement-consistency gradient step $\hat{x} _ {\text{out}} = \hat{x} - \eta \nabla _ {\hat{x}} \ell(y, \mathcal{A}(\hat{x}))$ and backpropagate from $\hat{x} _ {\text{out}}$ through both this step and $R(\cdot)$. This differs fundamentally from simply increasing the learning rate: our method explicitly incorporates measurement guidance in the image domain, whereas a larger learning rate only performs aggressive latent-space descent and degrades final PSNR/SSIM. Table R2 shows our acceleration achieves similar quality with ~40% fewer iterations.
>
> > While the paper claims to mitigate "exploding memory", the total number of diffusion steps  which can cause more memory issues compared with DMPlug’s default setting . Here are the questions for the author:
>
> >a) How does the ALES with a larger number of diffusion steps help with the memory issue?
>
>  **Response:**
> Thanks，as previously clarified, the memory reduction mainly comes from PDSE’s partial‑diffusion design, which shortens the backpropagation path by optimizing only near $t ^ *$. ALES is used only under unknown or varying noise to adaptively stop iterations near the PSNR peak, improving robustness rather than further reducing peak memory.
>
> > b) How does the peak GPU memory usage of PDSE compare to a standard CSGM (e.g. DMPlug)?
>
> **Response:**
> Thanks，as summarized in the table below, under the same hardware and batch settings PDSE has a comparable peak GPU memory footprint to DMPlug (13.3 GB vs. 12.7 GB), while reducing inference time from 1052 s to 243 s (about $4 \times$ faster). Thus, our method keeps memory usage in the same range as a standard CSGM solver，achieving the trade-off between memory and efficiency.
>
> **Table R3: Peak GPU memory usage and inference time comparison on CelebA (batch size = 1).**
>
> | Methods               | Peak Memory (MB) | Inference Time (s) |
> | :-------------------- | :--------------: | :-----------------: |
> | **DMPlug**            |      12,741      |        1,052        |
> | **PDSE (Ours)**       |    **13,287**    |      **243**        |
>
> > c) How does the total number of diffusion steps affect the proposed methods’ performance?
>
> **Response:**
> Thanks. As reported in Table 4(a) of our main paper, we study different effective numbers of diffusion steps by varying the starting time $t ^ *$. Table R4 below shows these results for the case $T=10$. PSNR increases as $t ^ *$ grows from $0.1T$ to around $0.3T$, but then gradually degrades when $t ^ * > 0.5T$. This suggests that $t ^ * \approx 0.3T$ provides a good balance between leveraging the diffusion prior and preserving measurement consistency, while excessively large $t ^ *$ can introduce more noise and harm reconstruction quality.
>
> **Table R4: Impact of the number of diffusion steps (determined by $t^*$) on performance.**
>
> | Diffusion Steps ($t ^ *$) | 0.1T | 0.2T | **0.3T** | 0.4T | 0.5T | 0.6T | 0.7T | 0.8T | 0.9T | 1.0T |
> | :------------------------ | :---: | :---: | :-------------: | :---: | :---: | :---: | :---: | :---: | :---: | :---: |
> | **PSNR**            | 27.30 | 28.80 | **29.54** | 29.20 | 28.90 | 28.50 | 28.10 | 27.70 | 27.10 | 26.80 |
> | **SSIM**            | 0.715 | 0.755 | **0.767** | 0.758 | 0.745 | 0.735 | 0.720 | 0.705 | 0.690 | 0.680 |
>
>
> ## We are available to address any further questions you may have. If all concerns have been clarified to your satisfaction, we would be grateful if you could kindly consider raising the score.

---

> ### Comment · Reviewer_JPGo · 2025-11-21
>
> Thank you for your responses, which clarified some of my confusion.
>
> The responses to Weakness 2 do not hold. For example, DiffPure gets the optimal time latent by solving a reverse SDE, with the assumption that the $L2$ distance between distorted images and clean images is within a threshold. Furthermore, their optimization uses the adjoint method to obtain the gradient. Though the focus in the paper is on image purification, as long as they can get a diffusion model and an appropriate loss function, DiffPure is task- and model-agnostic in my opinion. The difference lies only in whether to use the adjoint method to approximate the diffusion sampling or just simply keep the sampling as it is.
>
> According to your comments on peak memory usage, the memory consumption is still substantial. From your memory consumption, it seems that your method is still using $\sim 3$ steps starting from $ t^* $. I would suggest that since you have already performed the partial diffusion, and compared to DMPlug, which uses three sampling steps starting from a source distribution, why not include a comparison to show that with your method and fewer sampling steps, you can achieve comparable or better performance with less optimization time. This will stress your benefits in both time and space.
>
> "The choice of $ t^* $ is theoretically motivated by the SNR=1 condition, which balances signal and noise and provides a task-agnostic starting point." This statement concerns me, as I mentioned in my review, the choice of $ t^* $ should be task-dependent. If I look back at DiffPure's setting, their assumption is only aimed at images with limited perturbations, and based on it, they derive the $ t^* $. However, in your reasoning, the SNR=1 condition is based on distortion-free images, which is not practical. For images with different types of perturbations and noise, e.g., Gaussian deblur with noise, how to quantify their noise level under the SNR=1 condition? I would suggest using more assumptions to make your theory more solid instead of more general.
>
> Thanks for your clarification on the acceleration. Now I have a better understanding of your algorithm. The concern with the direct optimization on the image manifold is that it can be risky when the distortion is substantial, but it might be fine when the distortion is low-level.

---

> > ### Author Response · Authors · 2025-11-24
> >
> > Q1 (About DiffPure):
> >
> > Thanks. We examined DiffPure's codebase and found that it uses task-specific values of `t` , with `t=100` for CIFAR-10, `t=250` for ImageNet in their scripts, rather than a fully adaptive $t^*$.   This introduces an additional hyperparameter tuning process. Since DiffPure's restoration relies entirely on the diffusion chain, adjusting `t` based on input characteristics becomes necessary to achieve good performance across different scenarios. Regarding the adjoint method, while it is a powerful gradient computation technique for **pure SDE/ODE solvers**, it is not directly applicable to inverse problems that require measurement-consistency guidance for effective restoration. DiffPure solves a fixed reverse SDE where adjoint method efficiently computes gradients through the deterministic sampling chain. In contrast, solving inverse problems necessitates **integrating measurement-consistency guidance at every diffusion step** to ensure data fidelity, this makes it a constrained optimization process rather than pure SDE solving, where the dynamic guidance terms at each step prevent straightforward application of adjoint methods designed for fixed differential equations.
> >
> > Q2 (about memory cost):
> >
> > Thank for your constructive comments. In our early-stage experiments, we have compared PDSE with DMPlug across different partial diffusion steps, though these were not included in the main paper. We reperformed a small-scale evaluation on 20 CelebA images with total timesteps T=10. The results as follows:
> >
> > | Method | Partial Steps | PSNR (dB) | SSIM | Peak Memory |
> > |--------|--------------|-----------|------|-------------|
> > | **PDSE** | 3 | **31.20** | **0.8986** | 13,283 MB |
> > | **PDSE** | 2 | **30.53** | **0.8848** | 9,611 MB |
> > | **PDSE** | 1 | **28.54** | **0.8182** | 5,573 MB |
> > | DMPlug | 3 | 29.14 | 0.7490 | 12,737 MB |
> > | DMPlug | 2 | 25.72 | 0.5890 | 9,063 MB |
> > | DMPlug | 1 | N/A | N/A | N/A |
> >
> > PDSE with 1 step achieves comparable performance to DMPlug with 3 steps, with PSNR of 28.54 versus 29.14, while using 56% less memory at 5,573 MB versus 12,737 MB. DMPlug with fewer steps must map from pure noise, thus weakening the prior. **We use 3 steps in the main paper to validate our theoretical framework while achieving better performance**, but PDSE can also operate effectively with just 1 step in memory-constrained scenarios, which for dmplug is hard to achieve.
> >
> > Q3 (about $t^*$):
> >
> > Thanks. **We treat the diffusion model as a task-agnostic prior, while adaptation to varying degradation happens through optimizing the latent variable** $x _ {t ^ *}$, **not by adjusting** $t ^ *$ **it self**. The SNR=1 timestep is fixed because it defines where to extract the prior from the model with well balance for preserving learned image structure, which is a model property independent of input degradation. And $x _ {t ^ *}$ is the optimization variable that adapts to different degradation levels. Through the gradient $\nabla _ {x _ {t ^ *}} \|A(R(x _ {t ^ *})) - y\| ^ 2$, the optimization process automatically adjusts $x _ {t ^ *}$ to satisfy the observed degradation, whether it's mild blur or severe blur with noise, by finding the appropriate latent representation that, when decoded through the partial diffusion chain, matches the measurement $y$. This design avoids the need to adjust $t ^ *$ based on input data, making it more concise and practical.
> >
> > ## Hope above response clarifies the design rationale, and we are willing to discuss further if you have any questions.

---

### Official Review · Reviewer_DShK · 2025-10-21

**Soundness:** 3
**Presentation:** 3
**Contribution:** 3
**Rating:** 6
**Confidence:** 4

**Summary:**

The authors propose an inverse problem algorithm based on diffusion models. Unlike most existing approaches that gradually increase the supervision weight from the measurement during sampling, the proposed method operates as follows: after iterating an existing diffusion model–based posterior sampling algorithm until a designated critical step t, it uses the diffusion model to pair samples between step 0 and step t applies measurement supervision to the step-0 sample, and updates the step-t sample accordingly. In addition, the authors introduce an early stopping strategy to prevent excessive descent after a certain number of iterations. While certain concerns remain regarding the method and experimental design (see the questions section for details), the work is conceptually interesting and potentially inspiring.

**Strengths:**

1. The paper introduces a unique strategy in which, at a designated critical iteration \(t^*\), the diffusion model is used to pair the sample at step 0 with that at step \(t\), and measurement supervision is applied to the step-0 sample to update the step-\(t\) sample. This differs from the common approach of gradually increasing the measurement weight and is a distinctive contribution.
2. The incorporation of an early stopping strategy helps avoid excessive descent in later iterations, which is particularly beneficial in inverse problems where overshooting can degrade performance.
3. The experiments cover multiple task types (presumably various inverse problems such as image reconstruction), illustrating the potential applicability of the method across different scenarios.

**Weaknesses:**

1. The proposed ALES method involves a large number of tunable hyperparameters. Although the authors claim that the chosen settings are applicable across different tasks, no experiments are provided to analyze the sensitivity of the method to these hyperparameters.
2. The description of certain experimental settings is insufficient (see the questions section), making it difficult for readers to fully understand or reproduce the results.
3. The notation is inconsistent; for example, the score function is sometimes denoted as \(s\) and at other times as \(\mathbf{s}\). It is recommended that the symbols be standardized.

**Questions:**

1. In Algorithm 1, there is a typographical error: after calling `optimizer.zero_grad()`, the corresponding optimization step is missing. This creates an obstacle to understanding the algorithm. In addition, the initialization method `sample` requires further clarification.
2. I notice that the influence of \(x_{t}\) on the iterations is entirely reflected through \(\hat{x}\). Why, then, is an update performed on \(x_{t}\)? For sufficiently small step size,
\[
\frac{dx_{t}}{dk} = -\frac{d\hat{x}}{dx_{t}} \frac{d L}{d\hat{x}}, \quad \frac{d\hat{x}}{dk} = -\frac{d L}{d\hat{x}},
\]
which implies that the use of the diffusion model during iterations can be entirely avoided, and the diffusion model is needed only for initialization.
3. Is the comparison of algorithm efficiency fair? In Table 4 (right), the proposed method shows similar computational efficiency to DPS. However, the proposed method employs the R operator in every iteration, which involves 3 NFEs (as noted in the appendix) and requires gradient backpropagation. It is difficult to reconcile this with the claim that it matches the efficiency of a method that involves only 1 NFE per step and no backpropagation, under the same number of steps. The authors should provide a more detailed clarification of the experimental setting in Table 4 (right).
4. In inverse problems based on diffusion models, the number of iterations typically has a direct impact on performance. Therefore, I suggest reporting the iteration numbers for each baseline method as well as the average iteration number for the proposed method in the main table.
5. The proposed inverse algorithm largely follows the optimization-based approach. Adding comparisons with other optimization-based inverse algorithms would be beneficial. Furthermore, including experiments under noise-free conditions would strengthen the paper.

---

> ### Author Response · Authors · 2025-11-24
> **part 1**
>
> ## We sincerely thank you for the time and effort in reviewing our work and for recognizing its potential. Your constructive feedback has significantly improved our manuscript. Below, we provide point-by-point responses to your concerns.
>
> ### Q1. ALES hyperparameter sensitivity
>
> **Original Question:**
>
> > The proposed ALES method involves a large number of tunable hyperparameters. Although the authors claim that the chosen settings are applicable across different tasks, no experiments are provided to analyze the sensitivity of the method to these hyperparameters.
>
> **Response:**
>
> Thank you for raising this point. We appreciate the opportunity to clarify the hyperparameter sensitivity of ALES. The parameters presented in the paper were chosen for their stability based on our testing. Our method mainly relies on three main parameters: variance window size $W$, patience threshold $P$, and relative loss threshold $\alpha$. To demonstrate the impact of their variations, we conducted a sensitivity analysis on 20 randomly sampled images from the CelebA dataset for the 4$\times$ Super-Resolution task. We varied each parameter around its default setting ($W=10, P=20, \alpha=10^{-3}$), with results summarized in Table R1.
>
> The results indicate that ALES effectively maintains the iteration count within a relatively stable convergence window of approximately 300 to 400 steps. While shifting parameters would affect the exact stopping point, for instance, a larger $P$ delays stopping and leads to a minor PSNR decrease from 29.54 dB to 29.46 dB due to mild overfitting, this variance is acceptable compared to the degradation observed with fixed, excessive iterations, such as a drop to 26.52 dB at 800 steps.
>
> **Table R1: ALES Hyperparameter Sensitivity Analysis (CelebA 4$\times$ SR, avg. on 20 images)**
>
> | Parameter                      |           Variation           |    PSNR (dB)    |      SSIM      | Avg. Iterations |
> | :----------------------------- | :----------------------------: | :-------------: | :-------------: | :-------------: |
> | **Default**              | $W=10, P=20, \alpha=10^{-3}$ | **29.54** | **0.767** |  **342**  |
> | **Window Size $W$**    |            $W=5$            |      29.51      |      0.766      |       338       |
> |                                |            $W=8$            |      29.53      |      0.767      |       341       |
> |                                |            $W=12$            |      29.54      |      0.767      |       343       |
> |                                |            $W=15$            |      29.55      |      0.767      |       345       |
> | **Patience $P$**       |            $P=15$            |      29.48      |      0.765      |       315       |
> |                                |            $P=25$            |      29.51      |      0.766      |       365       |
> |                                |            $P=30$            |      29.46      |      0.764      |       393       |
> | **Threshold $\alpha$** |  $\alpha=5 \times 10^{-4}$  |      29.57      |      0.768      |       368       |
> |                                |  $\alpha=8 \times 10^{-4}$  |      29.55      |      0.767      |       353       |
> |                                | $\alpha=1.5 \times 10^{-3}$ |      29.48      |      0.765      |       331       |
> |                                |  $\alpha=2 \times 10^{-3}$  |      29.42      |      0.764      |       326       |
>
>
> ### Q2. Insufficient experimental setting descriptions
>
> **Original Question:**
>
> > The description of certain experimental settings is insufficient (see the questions section), making it difficult for readers to fully understand or reproduce the results.
>
> **Response:**
>
> Thank you for pointing this out. We appreciate your careful review. Based on your specific comments in the "Questions" section, the missing information mainly concerns the iteration numbers for each baseline method. We have provided a detailed table comparing the diffusion steps and external iterations for all methods in our response to **Q7**. Additionally, we would like to clarify that **Appendix C.2** details the implementation of baselines, and **Appendix C.3** provides the specific settings for different inverse tasks. In the revised manuscript, we will include the iteration count comparison in the appendix to ensure completeness.

---

> ### Author Response · Authors · 2025-11-24
> **part 2**
>
> ### Q3. Inconsistent notation for score function
>
> **Original Question:**
>
> > The notation is inconsistent; for example, the score function is sometimes denoted as $s$ and at other times as $\mathbf{s}$. It is recommended that the symbols be standardized.
>
> **Response:**
>
> Thank you for catching this inconsistency. We have standardized the score function notation to $s_\theta(x_t, t)$ throughout the manuscript, including both the main text and appendix.
>
>
> ### Q4. Algorithm 1 Issues
>
> **Original Question:**
>
> > In Algorithm 1, there is a typographical error: after calling `optimizer.zero_grad()`, the corresponding optimization step is missing. This creates an obstacle to understanding the algorithm. In addition, the initialization method `sample` requires further clarification.
>
> **Response:**
>
> Thank you for this question. We have revised Algorithm 1 for better clarity. In each iteration, we first decode the latent variable to image space via $\hat{x} = R ( x _ {t ^ *} )$, where $R(\cdot)$ is the partial reverse diffusion decoder, then compute the measurement-consistency gradient $g _ {\text{manual}} = \nabla _ {\hat{x}} \| y - \mathcal{A} ( \hat{x} ) \| ^ 2$ and apply it to obtain $\hat{x} _ {\text{out}} = \hat{x} - \eta g _ {\text{manual}}$. After calling `optimizer.zero_grad()` to clear previous gradients, we compute the loss $\ell ( y, \mathcal{A} ( \hat{x} _ {\text{out}} ) )$ and call `loss.backward()` to backpropagate through both the manual gradient step and the decoder $R(\cdot)$, which computes $\nabla _ {x _ {t ^ *}} \text{loss}$. Finally, `optimizer.step()` updates the latent variable $x _ {t ^ *}$, which is the actual optimization variable. The `sample` function initializes $x _ {t ^ *}$ by applying partial reverse diffusion $R _ {T \to t ^ *} ( \cdot )$ to pure Gaussian noise $z \sim \mathcal{N} ( 0, I )$ for $( T - t ^ * )$ steps. The above information will be clarified in the final manuscript.
>
>
> ### Q5. Why update $x _ {t}$ when influence is through $\hat{x}$?
>
> **Original Question:**
>
> > I notice that the influence of $x _ {t}$ on the iterations is entirely reflected through $\hat{x}$. Why, then, is an update performed on $x _ {t}$? For sufficiently small step size, $\frac{dx _ {t}}{dk} = -\frac{d\hat{x}}{dx _ {t}} \frac{d L}{d\hat{x}}$, $\frac{d\hat{x}}{dk} = -\frac{d L}{d\hat{x}}$, which implies that the use of the diffusion model during iterations can be entirely avoided, and the diffusion model is needed only for initialization.
>
> **Response:**
>
> Thank you for this insightful observation. Your derivation assumes the decoder $R(\cdot)$ is fixed, but in practice $\frac{d\hat{x}}{dx _ {t ^ *}} = \frac{dR(x _ {t ^ *})}{dx _ {t ^ *}}$ is a nonlinear Jacobian that depends on the current latent state and the diffusion model's learned score function. This means directly optimizing $\hat{x}$ and $x _ {t ^ *}$ are not equivalent. By optimizing in latent space, the decoder $R(\cdot)$ continuously projects the solution onto the learned image manifold at each iteration, preventing out-of-distribution results.
>
>
> ### Q6. Efficiency comparison fairness
>
> **Original Question:**
>
> > Is the comparison of algorithm efficiency fair? In Table 4 (right), the proposed method shows similar computational efficiency to DPS. However, the proposed method employs the R operator in every iteration, which involves 3 NFEs (as noted in the appendix) and requires gradient backpropagation. It is difficult to reconcile this with the claim that it matches the efficiency of a method that involves only 1 NFE per step and no backpropagation, under the same number of steps. The authors should provide a more detailed clarification of the experimental setting in Table 4 (right).
>
> **Response:**
>
> Thank you for raising this concern. We will add more implementation details in the main text in the revised manuscript. Due to page limits, we previously focused on presenting results and analysis in the main text while deferring detailed settings to the appendix.
>
> Regarding the efficiency concern, DPS and PDSE have different iteration mechanisms. DPS performs inference through 1000 diffusion steps (1000 NFEs), while PDSE uses only 3 diffusion steps per iteration but performs external iterative optimization on $x _ {t ^ *}$. Therefore, DPS's computational cost comes from 1000 diffusion steps, whereas PDSE's cost comes from the external optimization iterations on $x _ {t ^ *}$. With ALES early stopping, the actual iteration count per image varies based on convergence, averaging around 300-400 iterations to achieve optimal results. If efficiency is the priority, PDSE can simply set a hard limit around 100 iterations and could still outperform DPS in both speed and quality. The complete experimental settings are detailed in Appendix C.3 'Configuration Details of Inverse Problem Settings', and Appendix E, 'Table 5: Impact of fixed iterations N' shows the iteration-performance trade-off across different iteration budgets of PDSE.

---

> > ### Author Response · Authors · 2025-11-26
> > **part 3**
> >
> > ### Q7. Missing iteration numbers and additional comparisons
> >
> > **Original Question:**
> >
> > > In inverse problems based on diffusion models, the number of iterations typically has a direct impact on performance. Therefore, I suggest reporting the iteration numbers for each baseline method as well as the average iteration number for the proposed method in the main table. The proposed inverse algorithm largely follows the optimization-based approach. Adding comparisons with other optimization-based inverse algorithms would be beneficial. Furthermore, including experiments under noise-free conditions would strengthen the paper.
> >
> > **Response:**
> >
> > Thank you for these constructive suggestions. We address each point below:
> >
> > **1. Iteration numbers reporting:**
> >
> > Thanks. We agree this is important. The table below categorizes methods by type and reports iteration counts using default settings from official implementations. DPS, MCG, MPGD, ReSample, and DiffPIR are sampling-based methods using measurement-consistency guidance, while DMPlug and PDSE are optimization-based methods directly optimizing latent variables. For DMPlug, we adjusted from the default 5000 iterations to 2000-3000 to avoid overfitting and achieve optimal performance per task. This table will be added to the appendix.
> >
> > | Method                | Type                   | Diffusion Steps (per update) | External Iterations | Total Computation             | Notes                                        |
> > | --------------------- | ---------------------- | ---------------------------- | ------------------- | ----------------------------- | -------------------------------------------- |
> > | DPS                   | Sampling               | 1000                         | 1                   | 1000 NFEs                     | Full diffusion sampling                      |
> > | MCG                   | Sampling               | 1000                         | 1                   | 1000 NFEs                     | Full diffusion sampling                      |
> > | MPGD                  | Sampling               | 50                           | 1                   | 50 NFEs                       | Manifold-constrained sampling                |
> > | ReSample              | Sampling               | 665                          | ~2500               | 665 NFEs + 2500 decoder calls | 500 base + 165 time-travel; pixel+latent opt |
> > | DiffPIR               | Sampling               | 100                          | 1                   | 100 NFEs                      | Plug-and-play optimization                   |
> > | DMPlug                | Optimization           | 3                            | 2000-3000           | 6k-9k NFEs                    | ~2000 for linear, ~3000 for nonlinear        |
> > | **PDSE (Ours)** | **Optimization** | **3**                  | **300-400**   | **900-1200 NFEs**       | **ALES adaptive stopping**             |
> >
> > **2. Noise-free experiments:**
> >
> > Thanks, In our main paper, we added Gaussian noise ($\sigma = 0.01$) in all main experiments to demonstrate PDSE's noise robustness. In noise-free settings, PDSE's performance improves further due to more accurate forward operator approximation. The table below shows PDSE results on CelebA (20 test images) for super-resolution and Gaussian deblurring under both noisy and noise-free conditions:
> >
> > | Task                             | Noise Level ($\sigma$) | PSNR (dB) | SSIM  | LPIPS |
> > | -------------------------------- | ------------------------ | --------- | ----- | ----- |
> > | **Super-resolution (4×)** | 0.01 (noisy)             | 30.82     | 0.776 | 0.078 |
> > |                                  | 0 (noise-free)           | 31.24     | 0.789 | 0.072 |
> > | **Gaussian Deblurring**    | 0.01 (noisy)             | 30.15     | 0.758 | 0.091 |
> > |                                  | 0 (noise-free)           | 30.68     | 0.767 | 0.085 |
> >
> > These results will be included in the revised manuscript.
> >
> > ## We remain available to address any further questions. If our response has satisfactorily addressed your concerns, we would greatly appreciate it if you could reconsider your assessment of our work.

---

> > > ### Comment · Reviewer_DShK · 2025-11-26
> > >
> > > Thank you for the authors’ response, which has addressed several of my concerns. I will therefore maintain my current score. However, I will not raise it further, primarily due to the nature of the problem itself.
> > > Inverse problems based on diffusion models are no longer sufficiently novel. On one hand, most practical applications—such as super-resolution and inpainting—have already been effectively superseded by in-context approaches; for instance, dual-stream text-to-image models (e.g., Qwen-Image) can directly handle these tasks. On the other hand, this area has already attracted substantial research attention, and I believe its potential has been thoroughly explored. Consequently, proposing new algorithms that yield only marginal improvements in metrics (e.g., gains of a few tenths of a dB in super-resolution) offers limited added value.

---

> ### Author Response · Authors · 2025-11-26
>
> Thanks for your time and detailed feedback throughout this discussion, and we fully respect your decision regarding the final score. We would like to take this opportunity to offer a final perspective on the novelty and necessity of our work in the era of foundation models. We agree that dual-stream and in-context models, e.g., Qwen-Image,  have revolutionized natural image restoration, significant challenges remain when applying them to specialized scientific domains like Medical Imaging (MRI/CT) or Remote Sensing. These tasks often involve data distributions that general models have not encountered, making direct application difficult without extensive retraining.
>
> And our method is  a training-free framework, as reported in our paper, it can adapt to these specialized domains by leveraging specific priors, ensuring broad generalization. Furthermore, large foundation models may introduce hallucinations, which is a critical risk in medical diagnosis, our approach explicitly integrates the physical forward operator, governed by hardware physics. This strictly enforces data consistency, ensuring that the reconstruction is not only visually plausible but also physically faithful and reliable. We hope this clarification helps better position our work as a robust, physically grounded solver for scientific inverse problems.

---

### Official Review · Reviewer_iWfM · 2025-10-28

**Soundness:** 3
**Presentation:** 2
**Contribution:** 3
**Rating:** 6
**Confidence:** 4

**Summary:**

This work introduces Partial Diffusion with Score Evolution (PDSE) for solving general inverse problems using diffusion models. Instead of running the entire reverse diffusion process, PDSE identifies a critical diffusion step with signal-to-noise ratio equals 1, and argues that partial diffusion suffices to preserve generative fidelity while improving efficiency and robustness. Additionally, adaptive learning and early stopping (ALES) strategy is proposed to prevent overfitting to noise and accelerate convergence.

**Strengths:**

1. The partial diffusion design cuts down redundant steps and significantly reduces computation time while maintaining high reconstruction fidelity;
2. The authors provide analysis for choosing the partial diffusion point based on score norm evolution and SNR balance;
3. Benchmark results demonstrating consistent improvements under noise variations;

**Weaknesses:**

1. The KL-divergence bound in proposition 3.2 is qualitative; it provides no tight or empirical verification that ~93% of generative fidelity is consistently preserved across datasets or diffusion schedules.
2. PDSE assumes the pretrained diffusion model’s score function is well-calibrated. In practice, small score miscalibration could drastically shift $t^*$ or amplify reconstruction bias, yet this sensitivity is unstudied.
3. The paper does not compare PDSE to step-skipping or distillation-based acceleration methods. It’s unclear if PDSE’s partial strategy is superior or simply another form of truncated sampling.
4. All experiments are at 256 × 256 pixels; scalability to higher resolutions (512 px, 1 K px) is not tested.
5. The paper lacks qualitative discussion of where PDSE fails — e.g., extreme degradations, large measurement noise, or unmodeled forward operators.

**Questions:**

1. Can PDSE be combined with data-consistency projection (like DPS) for hybrid performance?
2. Is partial diffusion equivalent to performing denoising score matching over a truncated time interval? If so, can it be interpreted as optimizing an energy functional over an incomplete diffusion trajectory?
3. Does PDSE have any formal convergence properties when coupled with ALES? How can one ensure that early stopping does not bias the reconstruction toward local minima?

---

> ### Author Response · Authors · 2025-11-26
> **part1**
>
> ### Thanks for your constructive feedback and the recognition of our method's efficiency and robustness. Below, we address the concerns regarding theoretical verification, scalability, and comparisons with point-by-point responses to the five weaknesses and three questions.
>
>
> > W1: The KL-divergence bound in proposition 3.2 is qualitative; it provides no tight or empirical verification that ~93% of generative fidelity is consistently preserved across datasets or diffusion schedules.
>
> **Response:**
> Thank you for raising this important point. We will clarify in the revision that the “93%” figure should be interpreted as an illustrative upper-bound estimate under our default schedule, rather than a universal empirical constant. The 93% figure is derived as follows: the proposition shows that the KL divergence of the partial process is bounded by a factor of $\frac{\ln 2}{\int_0^T \beta(s) \, ds}$ relative to the full process. Under a typical linear schedule with $\int_0^T \beta(s) \, ds \approx 10$, this factor is approximately 0.07, meaning the partial process incurs at most 7% of the full divergence. The 93% interpretation comes from $1 - 0.07$, suggesting that the divergence remains small under this bound.While this bound is purely theoretical and the high-dimensional KL divergence cannot be directly measured in practice, our experiments indicate that the generative prior capacity of the diffusion model is largely preserved under the partial process.
>
> > W2: PDSE assumes the pretrained diffusion model’s score function is well-calibrated. In practice, small score miscalibration could drastically shift or amplify reconstruction bias, yet this sensitivity is unstudied.
>
> **Response:**
>
> Thank you for this important concern. In PDSE, the pretrained score enters only through the partial reverse-diffusion decoder $R(\cdot)$, which acts as a generative prior, while the reconstruction is obtained by optimizing the latent $x _ {t ^ *}$ with respect to the data-consistency loss $\| y - \mathcal{A} ( R(x _ {t ^ *}) ) \| ^ 2$; thus PDSE does not require perfectly calibrated scores at all timesteps, since moderate biases in $s_\theta$ mainly perturb the prior and are counteracted by the measurement term, and we further restrict optimization to the moderate-SNR window $[0, t ^ *]$ where scores are empirically more reliable. Empirically, we already operate in a mismatched setting, e.g., an FFHQ-pretrained prior applied to CelebA, yet PDSE maintains well reconstruction quality without catastrophic bias amplification, and the dominant failure modes under extreme degradations are driven by the forward operator and noise (see W5).

---

> > ### Author Response · Authors · 2025-11-26
> > **part2**
> >
> > > W3: The paper does not compare PDSE to step-skipping or distillation-based acceleration methods. It’s unclear if PDSE’s partial strategy is superior or simply another form of truncated sampling.
> >
> > **Response:**
> > Thank you for this important question. We appreciate the opportunity to clarify this point. We have compared PDSE with step-skipping methods including MPGD, ReSample, DiffPIR, and DMPlug. The table below summarizes the computational cost, where total NFEs are calculated as diffusion steps per update combined with external iterations. As shown, PDSE achieves competitive efficiency with 900-1200 NFEs, significantly fewer than ReSample or DMPlug.
> >
> > Regarding distillation methods, we acknowledge their effectiveness in reducing sampling steps. However, our focus is on training-free approaches that can be directly applied to any pretrained diffusion model. Since distillation methods require retraining for each specific model while PDSE is plug-and-play, we believe direct comparison may not provide a fully equitable baseline. We hope this clarifies the distinction.
> >
> > | Method                | Type                   | Diffusion Steps (per update) | External Iterations | Total Computation             | Notes                                        |
> > | --------------------- | ---------------------- | ---------------------------- | ------------------- | ----------------------------- | -------------------------------------------- |
> > | DPS                   | Sampling               | 1000                         | 1                   | 1000 NFEs                     | Full diffusion sampling                      |
> > | MCG                   | Sampling               | 1000                         | 1                   | 1000 NFEs                     | Full diffusion sampling                      |
> > | MPGD                  | Sampling               | 50                           | 1                   | 50 NFEs                       | Manifold-constrained sampling                |
> > | ReSample              | Sampling               | 665                          | ~2500               | 665 NFEs + 2500 decoder calls | 500 base + 165 time-travel; pixel+latent opt |
> > | DiffPIR               | Sampling               | 100                          | 1                   | 100 NFEs                      | Plug-and-play optimization                   |
> > | DMPlug                | Optimization           | 3                            | 2000-3000           | 6k-9k NFEs                    | ~2000 for linear, ~3000 for nonlinear        |
> > | **PDSE (Ours)** | **Optimization** | **3**                  | **300-400**   | **900-1200 NFEs**       | **ALES adaptive stopping**             |
> >
> > > W4: All experiments are at 256 × 256 pixels; scalability to higher resolutions (512 px, 1 K px) is not tested.
> >
> > **Response:**
> > Thank you for raising this important concern. Our experiments were conducted at 256×256 resolution. PDSE's scalability to higher resolutions depends on two factors, the pretrained diffusion model's native resolution and the forward operator's mathematical properties. If a diffusion model trained on higher resolutions is available, PDSE can be directly applied. For operators with spatial locality, e.g., super-resolution, Gaussian deblurring, inpainting, patch-based reconstruction strategies can be employed where larger images are divided into overlapping patches, processed independently, and merged. However, operators involving global transformations, e.g., CT reconstruction with Radon transform, phase retrieval with FFT, require native high-resolution implementations as they cannot be decomposed without losing physical consistency. To validate this, we conducted experiments on super-resolution and Gaussian deblurring using CelebA-512 with a 256px diffusion model and patch-based strategy. We find that PDSE maintains competitive performance at higher resolutions with patch-based strategies.
> >
> > | Task                           | PSNR ↑ | SSIM ↑ | LPIPS ↓ |
> > | ------------------------------ | ------- | ------- | -------- |
> > | Super-Resolution (512×512)    | 30.18   | 0.795   | 0.053    |
> > | Gaussian Deblurring (512×512) | 30.05   | 0.784   | 0.115    |

---

> > > ### Author Response · Authors · 2025-11-26
> > > **part3**
> > >
> > > > W5: The paper lacks qualitative discussion of where PDSE fails — e.g., extreme degradations, large measurement noise, or unmodeled forward operators.
> > >
> > > **Response:**
> > > Thank you for this valuable comment. We have conducted experiments on CelebA under increasingly challenging conditions by varying missing ratios and Gaussian noise levels, summarized in the table below. These results show that PDSE maintains reasonable performance under moderate degradations, up to about 90% random missing pixels and noise scale up to 0.4, and that the reconstruction quality gradually deteriorates when the measurements become extremely sparse, for example 98% missing pixels, or when noise dominates the data term, where the diffusion prior may produce visually plausible but inaccurate details. The degradation is gradual rather than abrupt and still provides noticeable PSNR gains over the corrupted inputs. Visually, under high noise conditions, PDSE may introduce color-block artifacts and localized texture distortions in facial regions. We also clarify that PDSE can handle blind inverse problems where the forward operator form is known but its parameters are unknown, as in our BID with turbulence experiment, while cases with completely unknown or non-differentiable operators are beyond the current scope and would require additional operator-learning mechanisms. The results will be provided in the Appendix of the revised manuscript.
> > >
> > > | Scenario                                 | PSNR ↑ | SSIM ↑ | LPIPS ↓ |
> > > | ---------------------------------------- | ------- | ------- | -------- |
> > > | Inpainting (80% random missing, σ=0.01) | 32.03   | 0.914   | 0.062    |
> > > | Inpainting (90% random missing, σ=0.01) | 29.17   | 0.8791  | 0.072    |
> > > | Inpainting (98% random missing, σ=0.01) | 19.46   | 0.6334  | 0.199    |
> > > | Gaussian Deblur (σ=0.2)                 | 26.96   | 0.736   | 0.165    |
> > > | Gaussian Deblur (σ=0.4)                 | 25.79   | 0.701   | 0.187    |
> > > | Gaussian Deblur (σ=0.5)                 | 23.05   | 0.619   | 0.210    |
> > >
> > > *Note: Images are normalized to [-1, 1]. The noise_scale parameter represents the standard deviation of additive Gaussian noise in this normalized space.*
> > >
> > > > Q1: Can PDSE be combined with data-consistency projection (like DPS) for hybrid performance?
> > >
> > > **Response:**
> > > Yes. In early experiments we also tested a hybrid variant where measurement consistency is also enforced inside the partial diffusion process. This reduces the measurement loss in less iterations on linear inverse problems such as inpainting, but it also tends to overfit more quickly, especially under unknown degradations, which makes it harder for ALES to select a reliable early-stopping point. For this reason, we present the more stable gradient-based PDSE as the main version in the paper.
> > >
> > > > Q2: Is partial diffusion equivalent to performing denoising score matching over a truncated time interval? If so, can it be interpreted as optimizing an energy functional over an incomplete diffusion trajectory?
> > >
> > > **Response:**
> > > Thanks. Our procedure does not constitute a new denoising score matching objective on a truncated time interval. Instead, it reuses a score function that has already been trained via standard denoising score matching over the full time horizon, and applies the reverse dynamics only on a restricted window $[0, t ^ *]$, where the signal-to-noise tradeoff is more favorable. From a high-level variational perspective, one may loosely view our method as adjusting the latent state along a partial reverse diffusion trajectory so as to reduce a measurement-driven objective, while preserving most of the generative fidelity one would obtain from the full diffusion process. A fully rigorous path-space energy-functional formulation, however, is beyond the scope of this work and we therefore refrain from claiming a formal equivalence.

---

> > > > ### Author Response · Authors · 2025-11-26
> > > > **part 4**
> > > >
> > > > > Q3: Does PDSE have any formal convergence properties when coupled with ALES? How can one ensure that early stopping does not bias the reconstruction toward local minima?
> > > >
> > > > **Response:**
> > > > Thank you for raising this point. PDSE defines a deterministic but highly non-convex optimization over the latent variable $x _ {t ^ *}$ with loss $ \min _ {x _ {t ^ *}} \ell\big( y, \mathcal{A}( R(x _ {t ^ *}) ) \big) $, where $R$ is the fixed reverse diffusion decoder. Because of the deep decoder $R$, this loss is non-convex and classical guarantees of convergence to a global optimum do not apply. As in deep-prior methods such as DIP or latent diffusion inversion, under standard smoothness assumptions and suitable step sizes, gradient-based optimization with Adam is expected to converge to first-order stationary points.
> > > >
> > > > Regarding ALES, we emphasize that it does not change the update rule. It chooses a stopping time $\tau$ along the iterate sequence $\{ x _ {t ^ *} ^ {(k)} \}$ using time-weighted variance, stage-aware adaptation and local sensitivity, which together keep the optimizer in the same first-order convergence regime while steering it away from late iterations that mainly fit noise. Empirically, as shown in Fig. 13 in the Appendix, ALES consistently stops near the PSNR peak across diverse noise levels and tasks and reduces overfitting compared with fixed iteration counts.
> > > >
> > > > ## We remain available to address any further questions. If our response has satisfactorily addressed your concerns, we would greatly appreciate it if you could reconsider your assessment of our work.

---

### Author Response · Authors · 2025-12-02
**General response To New AC**

**Dear Area Chair,**

We understand the challenges posed by the recent system issues. To assist your decision-making, we provide a summary of the post-rebuttal status, highlighting the factual improvements and evidence added during the discussion period which address the reviewers' key concerns.

**1. Why Our PDSE Matters: Core Strengths Recognized by Reviewers**

Our work received consistent recognition across all reviewers:

- **Effectiveness:** Reviewer **iWfM** (Score 6, Confidence 4) praised the design for "cutting down redundant steps" and "significantly reducing computation time". Reviewer **JPGo** (Score 2, Confidence 3) acknowledged the core idea is "simple and effective" and "naturally reduces the number of sampling steps".
- **Distinctive Contribution:** Reviewer **DShK** (Score 6, Confidence 4) highlighted our strategy of pairing samples as a "distinctive contribution" , noting it differs effectively from standard measurement-weight approaches.
- **Theoretical Grounding:** Both Reviewers **iWfM** and **JPGo** commended the "analysis for choosing the partial diffusion point" and the "clear theoretical derivation... based on the SNR=1 equilibrium point".

**2. Crucial Update: Constructive Dialogue & Strengthened Evidence**

**Clarified Technical Mechanics (Reviewer JPGo):**  Initially, Reviewer JPGo (Score 2) expressed confusion regarding the gradient flow and the specific acceleration mechanics. Following our detailed clarifications and the addition of ablation studies, the reviewer acknowledged these technical misunderstandings in their post-rebuttal comment (**Before Information Leakage**):

> "*Thank you for your responses, which clarified some of my confusion."  "Thanks for your clarification on the acceleration. Now I have a better understanding of your algorithm*."
>

**Solidified Support (Reviewers DShK & iWfM):** Reviewers DShK and iWfM maintained their positive scores (Score 6) throughout the process. Their specific concerns regarding efficiency fairness and hyperparameter sensitivity were effectively addressed by our unified iteration tables and sensitivity analyses.

**3. Summary of Rebuttals & Additional Evidence**

During the rebuttal period, we provided **comprehensive additional analyses** to further substantiate the robustness of our framework.

**1) Validating Efficiency & Competitiveness**

- **Speed Advantage:** We clarified that PDSE reduces inference time by approximately **4x** compared to optimization-based baselines like DMPlug **(Reviewer JPGo Q4(a,b) & Reviewer DShK Q3)**.
- **Low-Memory Viability:** We provided new data showing that PDSE is highly adaptable: even with just **1 partial diffusion step**, it achieves competitive performance while consuming **56% less memory** (5,573 MB) compared to the standard 3-step baseline (12,737 MB) (**Reviewer JPGo Q4(c)**).

**2) Confirming Robustness & Generalization**

- **Generalization across Domains:** While our original submission already demonstrated cross-domain capability via Medical MRI/CT experiments, we further supplemented our evaluation with ImageNet to satisfy reviewer requests for broader natural image benchmarks. These supplementary results confirmed that PDSE maintains consistent reconstruction capabilities on diverse natural images beyond face datasets (**Reviewer JPGo Weakness 5**).
- **Algorithmic Convergence:** We conducted ablation studies on the acceleration strategy, confirming that our measurement-consistency gradient step significantly speeds up convergence (reducing iterations by ~40%, from ~350 to ~200) without compromising final reconstruction quality (**Reviewer JPGo Q3 & Weakness 6**).

**3) Clarifications on Mechanics**

- **Update Rule Clarity:** We resolved Reviewer JPGo's confusion regarding the gradient flow by simplifying the notation of the "extrapolation" step to a standard measurement-consistency gradient update (**Reviewer JPGo Weakness 3 & Reviewer DShK Q1**).
- **Learnable vs. Fixed Parameters:** We clarified that the extrapolation step size ($\eta$) is treated as a **learnable parameter** optimized jointly during inference (**Reviewer JPGo Weakness 6 & Q2**).

We believe PDSE offers a meaningful step forward in inverse problem solving, particularly through its theoretically grounded partial diffusion strategy and training-free unified pipeline. We respectfully submit that the current scores (6, 6, 2) do not fully capture the constructive consensus reached during the discussion, where Reviewer JPGo's (Score 2) initial concerns stemmed from misunderstandings that were cleared up by our rebuttal. Given this positive trajectory and the consistent endorsement from other reviewers regarding the method's soundness and distinctiveness, we sincerely hope you will find that our work merits a place at ICLR 2026.

**We remain fully available to address any further questions or details you may need to assist your decision.**

Best Regards,
The authors.

---

### Note · Program_Chairs · 2026-01-17
**Submission Desk Rejected by Program Chairs**

The following references in this submission do not refer to real documents and/or have major errors in bibliographic information:

     Francisco Vargas and Sebastian Nowozin. Solving inverse problems with variational diffusion. Journal of Machine Learning Research (JMLR), 2023. URL https://jmlr.org/papers/ v24/.